



# 21st century estimates of mass loss rates from glaciers in the Gulf of Alaska and Canadian Archipelago using a GRACE constrained glacier model

Lavanya Ashokkumar[1] and Christopher Harig[1]

[1]Department of Geosciences, University of Arizona, Tucson, AZ 85721

**Correspondence:** Lavanya Ashokkumar (lashokkumar@email.arizona.edu)

**Abstract.** Ice mass loss rates from glaciers in the Gulf of Alaska and the Canadian Archipelago are expected to increase through the end of century in response to increasing temperatures. Here, we develop a new glacier model constrained by GRACE gravimetry observations for the period between 2002 and 2017. The high temporal and regional spatial resolution of GRACE mass balance estimates allows us to estimate regional glacier sensitivities to atmospheric changes, and account for

higher order of glacier dynamics. We use our regionally constrained models to extrapolate future mass loss under different climate emission scenarios. Generally our 21st century sea level estimates are at the high end compared to other studies. We find that the Gulf of Alaska exhibits the highest mass loss rates between -79 to -112 Gt yr$^{-1}$ between 2006 and 2100 under different scenarios, and displays the highest sensitivity to the specific scenario (RCP 2.6/4.5/8.5). Our estimates for Arctic Canada South are significantly higher than prior work (-57 to -85 Gt yr$^{-1}$) and are comparable to projected mass loss rates

from the Arctic Canada North (-63 to -101 Gt yr$^{-1}$).

## 1 Introduction

During the satellite instrumental era (1993–2009) mountain glaciers contributed approximately 24% of observed global mean sea level rise (IPCC, 2019). Amongst these glaciers, the Gulf of Alaska (GA) and Arctic Canada North (ACN) or Ellesmere and South (ACS) or Baffin regions are the primary hotspots of negative mass balance since the 1960s (Gardner et al., 2012,

2013; Harig and Simons, 2016; Cook et al., 2019; Wouters et al., 2019; Zemp et al., 2019). Between 1961 and 2016, the Gulf of Alaska region has contributed to a total mass loss of -3000 Gt or 8 mm sea-level rise with a consistent negative mass balance of -0.6 m.w.e.yr$^{-1}$ (Zemp et al., 2019). Glaciers and ice caps in the Arctic Canada North and South contributed -28.2 $\pm$ 11.5 Gt yr$^{-1}$ and -22.0 $\pm$ 4.5 yr$^{-1}$ over the last two decades (Noël et al., 2018). Recent studies indicate that there has been a sharp increase in the rates of mass loss over the last few years. From a combination of direct and geodetic observations, mass loss has

increased in the Gulf of Alaska (-73 $\pm$ 17 Gt yr$^{-1}$) and Arctic Canada North(-60 $\pm$ 84 Gt yr$^{-1}$) from 2006 to 2016 (Zemp et al., 2019). The results are consistent with gravimetry observations, which indicate an acceleration of -8 Gt yr$^{-2}$ for Arctic Canada North and -3 Gt yr$^{-2}$ between 2003 and 2013 (Luthcke et al., 2013; Harig and Simons, 2016). Significant rates of mass loss has been reported from these regions that rather constitute a small proportion ($\sim$10-15%) in the global glacier population. In the Gulf of Alaska, the melt is primarily driven by regional meteorological forcing in temperature and precipitation, along with



strong feedback from glacier hypsometry (Luthcke et al., 2008; Arendt et al., 2013; Larsen et al., 2015). In contrast, glaciers
     in the Canadian Archipelago are greatly influenced by the oceanic control of heat transport, warmer temperatures (+1.1 C) and
     surface albedo feedback from complex glaciers and ice caps (Lenaerts et al., 2013; Noël et al., 2018; Cook et al., 2019).

     Due to the increased response of glaciers to climate change, it is important to understand the future rates of mass loss
     from glaciers based on climate scenarios. Authors in the glacier modeling community construct parameterized mass-balance
models for individual glaciers constrained by direct contemporary observations and static regional mass balance, and then
     estimate how a glacier's volume will evolve with time from knowledge of atmospheric variables over that glacier. One of
     the drawbacks in many studies is the extrapolation of regional mass balance from about 255 direct observations to represent
     200,000 glaciers worldwide (Cogley, 2009). The temporal resolution of these observations are sparse, sometimes less than 5
     years and inconsistent over the time period from 1956 to present. Moreover, most of the direct measurements are spatially
concentrated in the Alps, Western Canada, the Canadian Archipelago, and Russia, with sparse measurements from glaciers
     in the South Hemisphere or from high orographic regions such as the High Mountain Asia or Andes. This explains some of
     the large uncertainties from regional mass balance extrapolation in glacier modeling (Radić and Hock, 2011; Marzeion et al.,
     2012; Giesen and Oerlemans, 2013).

     Alternatively, several studies have constrained a glacier model by regional remote sensing data and calibrated a glacier model
with gravity data from the Gravity Recovery and Climate Experiment (GRACE) (Arendt et al., 2013). Wahr et al. (2016) also
     used GRACE observations but additionally forced the model with re-analysis products, which correlate well with GRACE data
     (Arendt et al., 2002). In the study by Huss and Hock (2015), modelled mass balance from in-situ observations are matched
     with standardized regional estimates by Gardner et al. (2013) between 2003 and 2009. In this case, the model parameters or the
     process were unable to represent the mass balance from individual glaciers and the process could not explain the seasonality in
mass balance for the observational period. We build on the model of Wahr et al. (2016) and standardize our processing to align
     with the recent GlacierMIP intercomparison project (Hock et al., 2019).

     In this study, we use regional mass balance estimates from GRACE observations between 2002 and 2017 to calibrate mod-
     eled glacier mass balance, and estimate future rates of mass balance using climate model output. Using these observations we
     obtain unique regional sensitivity parameters that identify the regional response of glaciers. In this way, we aim to address the
uncertainties from extrapolation of direct observations and issues in volume-area scaling. This method has improved tempo-
     ral resolution and produces regional estimates of parameters and mass balance, which accounts for seasonality and regional
     dynamics. Like Arendt et al. (2013), our glacier model accounts for spatial variability in mass loss among glaciers within indi-
     vidual regions. We focus on the Gulf of Alaska and Canadian Archipelago which have abundant other studies for comparison,
     and present future mass loss projections through the end of the century for varying climate emissions scenarios.





## 2 Data and methods

### 2.1 Mass balance from GRACE

Monthly observations of Earth's gravity field provide estimates of changes in the surface water cycle, such as the changes in terrestrial water storage, ice mass storage of continental ice sheets and glaciers, ocean dynamics and currents, etc. which perturb the geoid at a spatial resolution of several hundred kilometers (Wahr et al., 1998; Swenson et al., 2003; Wahr et al., 2004). We calculate mass anomalies for the Gulf of Alaska and Canadian archipelago by processing time-variable GRACE gravity solutions (Tapley et al., 2004). We use 163 monthly GRACE RL06 Level-2 fields between 2002 and 2017, from the University of Texas Center for Space Research (CSR) released as spherical harmonic Stokes coefficients up to degree and order $L = 60$. Degree one coefficients for geocenter motion are included and are calculated as in Swenson et al. (2008). The GRACE field geopotential coefficients can be downloaded at https://doi.org/10.5067/GRGSM-20C06 (Bettadpur., 2018). Geopotential perturbations are converted to changes in surface mass density following the method of Wahr et al. (1998) by using load Love numbers that incorporate Earth's elastic response to loading. Degree two order zero ($C_{2,0}$) and degree two order one ($C_{2,1}$) coefficients are substituted from satellite laser ranging (SLR) (Cheng et al., 2013). We remove the viscous response to prior deglaciation using the glacial isostatic adjustment (GIA) model. However, our GIA model does not account for the post-glacial rebound due to Little ice age for glaciers in the Gulf of Alaska. This could involve a contribution of $\sim$3 Gtyr$^{-1}$ in the mass loss time series for GA. Near glaciers, we take into account changes in terrestrial water storage (TWS) by removing monthly estimates of TWS calculated from the Global Land Data Assimilation System 1.0 (Noah), except in areas covered by glaciers, where they are known to be unreliable (Arendt et al., 2013), in keeping with GRACE community practice (Shepherd et al., 2012). To delineate the influence of glaciers from TWS, a mask is created from RGI inventory at 0.5 ° spatial grid resolution.

The processed Level-2 gravity solutions contain a combination of random and systematic noise. Random errors arise with increasing spherical harmonic degree due to the nature of making observations at satellite altitude. Systematic errors exist in particular spectral domains (i.e. stripes) due to GRACE's particular orbit configuration and typical inversion strategy (Swenson et al., 2003; Swenson and Wahr, 2006; Wahr et al., 2006; Landerer and Swenson, 2012). Often these are removed or reduced through a combination of filtering techniques (e.g. Swenson and Wahr, 2006). Additionally, there is an estimation error that arises from estimating a local signal from a global continuous field (Swenson and Wahr, 2002). To address these sources of error, our method uses Slepian functions as an alternative basis set to spherical harmonics (described further in Section 2.2) (Simons and Dahlen, 2006). In brief, use of these basis functions at high latitude does not require additional smoothing or filtering of GRACE Level-2 products. We use the method of Wahr et al. (2006) to form a conservative estimate of the noise, which is then propagated into the formal data and trend uncertainty taking into account autocorrelation (Harig and Simons, 2016).

### 2.2 Slepian Functions

In order to measure ice mass changes from GRACE gravity data we use a method based on Slepian functions that can isolate those contributions from the global potential field and resolve them over our regions of interest. Slepian functions form an





alternate basis on the sphere to spherical harmonics and the functions maximize their energy within a specific region while being orthogonal both on the sphere and region. The localized basis is also sparse, improving signal-to-noise and estimation

on timeseries of each coefficient. Slepian functions have been used to obtain the mass loss rates from Greenland (Harig and Simons, 2012), Antarctica (Harig and Simons, 2015), and smaller regions such as Gulf of Alaska, Canadian Archipelago, and Iceland (Harig and Simons, 2016; von Hippel and Harig, 2019). The code to generate the mass loss timeseries are available at Harig and Simons (2015).

Here, we briefly summarize the construction of Slepian functions. When considered over partial sphere regions, $R$, traditional

Spherical Harmonics $Y_{lm}$ are no longer orthogonal; their integral products are no longer delta functions (as is the case over the whole sphere). Instead, the integral products up to a certain bandwidth $L$ form a matrix $\mathbf{D}$, the localization kernel, with significant off-diagonal power,

$$\int_R Y_{lm} Y_{l'm'} d\Omega = D_{lm,l'm'}. \tag{1}$$

We construct a new set of basis functions, Slepian functions, by taking the eigenvalue decomposition of $\mathbf{D}$ as

$$\sum_{l'=0}^{L} \sum_{m'=-l}^{l'} D_{lm,l'm'} g_{l'm'} = \lambda g_{lm}. \tag{2}$$

Slepian functions, $g$, are then the eigenfunctions of localization kernel. The functions are constructed as a combination of spherical harmonic functions. The eigenvalues, $\lambda$, vary between zero and one, and describe how well each eigenfunction is concentrated in the region. By considering only large eigenvalues that maximize the concentration ratio as

$$\lambda = \frac{\int_R g_\alpha^2(\hat{\mathbf{r}}) d\Omega}{\int_\Omega g_\alpha^2(\hat{\mathbf{r}}) d\Omega} = \text{maximum}, \tag{3}$$

we can create a new truncated basis set of only the most concentrated functions, which are orthogonal both on the whole sphere $\Omega$ and over the region $R$. The Shannon number,

$$N = (L+1)^2 \frac{A}{4\pi}, \tag{4}$$

where N describes the effective number of functions that can be concentrated within a region for a given bandwidth ($L$) and region size ($A$). Practically, this includes functions with eigenvalues $\lambda > 0.5$. We then use this truncated set of functions in

traditional signal estimation where

$$\hat{s}(\mathbf{r}) = \sum_{\alpha=1}^{N} \hat{s}_\alpha g_\alpha(\mathbf{r}), \text{ and } g = \sum_{lm}^{L} g_{lm} Y_{lm}. \tag{5}$$

The local signals in GRACE data are efficiently recovered by transforming spherical harmonics $Y_{lm}(\hat{\mathbf{r}})$ into Slepian functions $g_\alpha(\hat{\mathbf{r}})$ up to the Shannon number ($N$). This transformation reduces the number of basis functions needed to represent signals in a





region. For example, for the Gulf of Alaska region the original $(L+1)^2 = 3721$ functions are reduced to $N = 4$ functions. Thus,
the reduction in the number of functions used in signal estimation increases the signal-to-noise of each function. This method
has been shown to obtain higher spatial resolution information from the existing monthly Level-2 GRACE data products than
other methods, with clear estimates of noise uncertainty (Harig and Simons, 2012, 2016).

## 2.3  Glacier mass balance model from climate data

The mass balance model is developed from the use of input gridded climate reanalysis products, global glacier inventory from
the Randolph Glacier inventory (RGI) 6.0 (RGI et al., 2017) and initial assumptions in volume area scaling. The modeling pro-
cess is based on a temperature-index degree day calculation, and it broadly includes the following components: (a) computation
of ablation and accumulation from ERA-Interim temperature and precipitation for the observational period between 2002 and
2017, (b) inputs of hypsometry such as the glacier outlines, area and elevation for volume and mass balance calculations, (c)
glacier assumptions for the initial conditions of area and volume. First, we describe the details of the datasets used in modeling,
followed by the computation of regional mass balance.

We estimate monthly intervals of glacier mass balance from 2002 to 2017 using the temperature and precipitation products
from ERA-Interim (ERAI) reanalysis datasets (Dee et al., 2011; ECWMF., 2012). ERAI is a global gridded data assimilation
product, provided by the European Center for Medium-Range Weather Forecasts (ECMWF) at a spatial resolution of $0.7°$ from
1979 to 2018. We use ERAI geopotential temperature and ERAI precipitation data products in the calculation of ablation and
accumulation, respectively, described further below.

The RGI contains glacier information for 563 glaciers from the Gulf of Alaska North, 4000 glaciers in the Canada North and
7413 glaciers in the Arctic Canada South. The glacier outlines are provided at a spatial resolution of $0.5° \times 0.5°$. To maintain
the consistency between the RGI regional outlines and GRACE regions of interest, we include only the glaciers defined by
RGI regional boundaries.



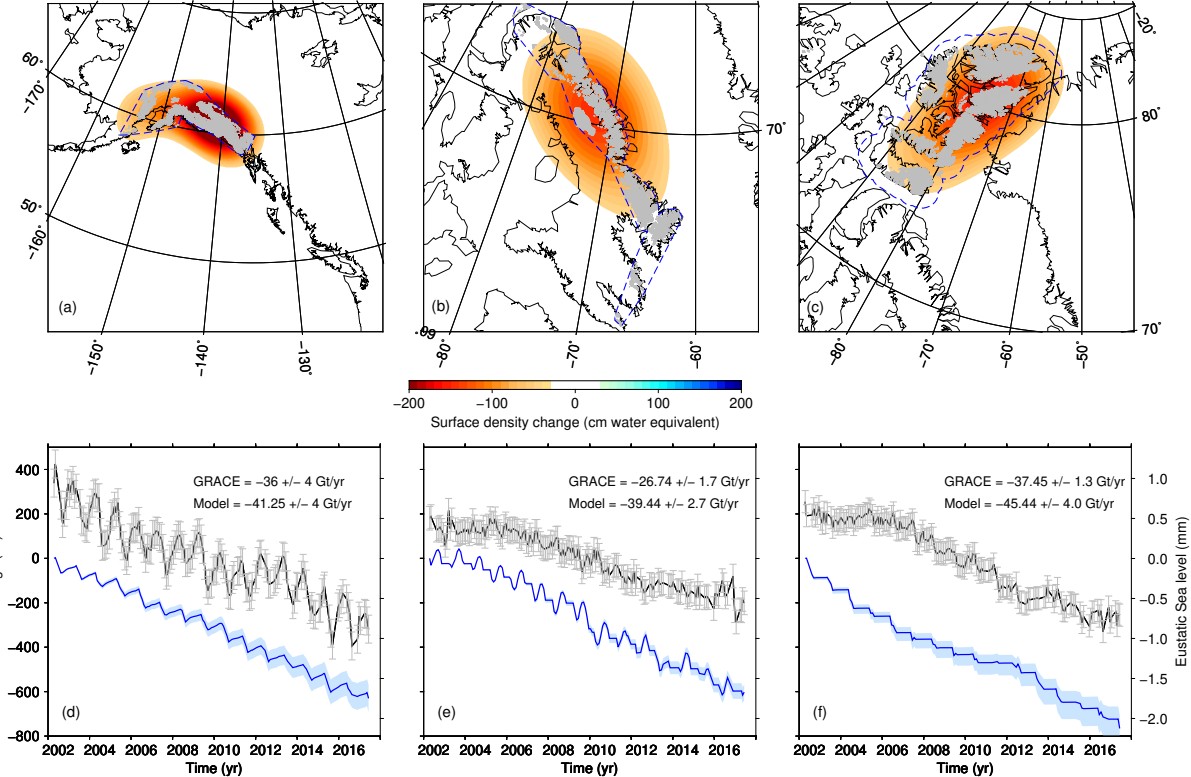

**Figure 1.** Map of total ice mass loss obtained from GRACE gravity data (top row, a–c) and comparison of GRACE derived total regional mass loss with modeled glacier mass balance (bottom row, d–f). The glacier inventory (RGI 6.0) is marked with filled light grey polygons. The black dashed lines mark the region of localization. Our regions of interest include Gulf of Alaska (a, d), Arctic Canada South (Baffin) (b, e), and Arctic Canada North (Ellesmere) (c, f) in the Arctic Canadian Archipelago. The corresponding timeseries of mass change from GRACE are indicated in black with monthly errorbars (d–f). The timeseries of mass change obtained from the glacier mass balance model for the three regions is indicated in blue.

For Gulf of Alaska, we consider glaciers from the Gulf of Alaska North region due to the regional difference in mass loss rates compared to the Gulf of Alaska South region (Arendt et al., 2002). RGI provides hypsometric details of glacier area at an elevation interval of 50 m. From this information, we compute the area elevation distribution of glaciers at every 50 m grid spacing. We compute the regional mass balance based on elevation bins and individual glaciers defined by the RGI.

We use the degree day approach to model the present and future mass balance for transient conditions of temperature and precipitation for glaciers in the Gulf of Alaska and Canadian Archipelago. For every glacierized cell of RGI at 0.5° spatial resolution, monthly rates of ablation and accumulation are calculated based on the relationship between local meteorological condition and glacier hypsometry (Radić and Hock, 2006, 2011). The mass balance ($M$) of a glacier is given as the sum of ablation ($A$), accumulation ($C$) and refreezing ($R$) as

$$M = A + C + R. \tag{6}$$





Here in our model, we ignore the effects of refreezing components from individual glaciers due to its minimal contribution in
the regional mass balance (Wahr et al., 2016).

We calculate ablation for each glacier, $A_{gl}$, from ERAI temperature at degree day ($DDF$, mm.w.e.d$^{-1}$C$^{-1}$), which is
typically the number of melt days above the threshold temperature $T_o$, to differentiate the solid and liquid precipitation, as

$$A_{gl}(t) = \left\{ \begin{array}{ll} 0, & \text{if } T(t) \leq T_o \\ DDF * (T(t) - T_o), & \text{if } T(t) \geq T_o \end{array} \right\}. \tag{7}$$

Surface temperature or temperature at the glacier, $T_{gl}$, is calculated from downscaling ERAI geopotential temperature based
on the lapse rate, $lp$, where $h$ is the mean elevation of an individual glacier, and $\Delta h$ is the average elevation of glaciers in a
region,

$$T_{gl}(h,t) = T(t) + lp * (h - \Delta h). \tag{8}$$

Accumulation is based on the precipitation, which is the amount of snowfall considered as ice. Like temperature, the pre-
cipitation, $P_{gl}$, is downscaled from convective precipitation $P(t)$ using a precipitation gradient $d_{prec}$, and average elevation of
glaciers in the region $\Delta p$,

$$P_{gl}(h,t) = P(t) * [1 + d_{prec} * (h - \Delta p)]. \tag{9}$$

There is a slight difference in the temperature and precipitation lapse rates because the precipitation is not accurately rep-
resented at high altitudes, especially from the climate reanalyses data products (Behrangi et al., 2016). Accumulation from
glaciers $C_{gl}(t)$ is then calculated as,

$$C_{gl}(t) = K_o P_{gl}, \tag{10}$$

where $K_o$ is the precipitation correction factor that is used to account for snowfall from all the glaciers.

While modeling the mass balance of individual glaciers in the RGI, the accumulation and ablation rates are computed by
matching each glacier to the nearest ERA-Interim grid cell. We ignore the effects of tidewater calving from Gulf of Alaska and
Canadian Archipelago since they contribute less to regional mass balance (Larsen et al., 2015). The mass balance is calculated
in two ways, based on elevation bins within the region of interest, and by considering each individual glacier. We find that
the computation of mass balance from glaciers is useful for model tuning, contrary to the elevation binned mass balance as in
previous glacier modeling studies (Huss and Hock, 2015). We also found that the results from these two methods does not vary
much.

**2.4   Model initialization, tuning, and constraints from GRACE data**

Our glacier model derived using ERAI dataset is highly sensitive to the primary parameters that include the degree-day factor
($DDF$), threshold temperature ($T_o$), precipitation correction factor ($K_o$) and less sensitive to the temperature lapse rate ($\Delta h$),
precipitation lapse rate ($\Delta p$), precipitation gradient ($d_{prec}$), and environmental lapse rate ($lp$) (Table 5). In order to constrain





our model to an observational record, we use the monthly timeseries of regional mass balance from GRACE (Section 2.1) to
compare against our monthly modeled glacier mass balance. We limit our model space to solve for three primary parameters,
DDF, $T_o$ and $K_o$ (bold in Table 5), and fix the remaining parameters to values used in previous studies (e.g. Wahr et al., 2016).
The primary parameters have greater sensitivity for controlling the modeled mass balance. The model is run for individual
glaciers within our region of interest, and it is integrated for all glaciers to obtain total mass balance. We then use a grid search
optimization using the parameter ranges listed in Equation (11) to solve for the combination that best minimizes the residual
sum of squares between modeled mass balance and GRACE observed mass balance, calculated as

$$\sum_{t=1}^{n}(GRACE(t) - M(t))^2, \text{with} \begin{bmatrix} DDF = 1,2,...6 \\ T_o = 270, 271, ...285 \\ K_o = 0.12, 0.24, ...1.10 \end{bmatrix} \tag{11}$$

One key difference between our model and prior work is that we calculate the optimal model parameters ($DDF$, $T_o$, and $K_o$)
in each region separately because we have separate regional GRACE mass balance timeseries.

The modeled mass balance explains nearly 87% to 88% of variance from the GRACE mass balance (depending on region,
see figure A3) using three parameters mentioned in Table 5. Even though the GRACE regional mass balance would not be
expected to represent the changes in any specific individual glacier, we compare our modeled mass balance with several
individual glaciers that have direct observations in the WGMS in Sections 3 and 4.3.

## 2.5  Future mass and volume rates

We use the optimized mass balance model developed using ERA Interim to predict future mass and volume loss rates until 2100.
Climate model data from CESM-LE (Kay et al., 2015), HadGEM-ES (Bellouin et al., 2011), and MRI-CGCM3 (Yukimoto
et al., 2012) was used for transient conditions of temperature and precipitation under several different emission scenarios.
Table 2 shows the resolution and details of climate model data. Prior to future projections, we adjusted the bias in temperature
$T_i(t)$ and precipitation $P_i(t)$ between ERA-Interim and each climate model using a delta approach (Radić and Hock, 2006) for
the baseline period between 2002 and 2017 as

$$T_i(t) = T_{i,c}(t) + (\overline{T}_{i,ERA} - \overline{T}_{i,c}), \text{ and} \tag{12}$$

$$P_i(t) = P_{i,c}(t) + (\frac{\overline{P}_{ERA}}{\overline{P}_c}), \tag{13}$$

where $i = 1, 2, ...12$ for each month and $c$ refer to the model/emission scenario combination. The spatial resolution of grid
cells near the poles was accounted in the computation of mean temperature and precipitation within our region of interest.
We found that the temperature pattern from both the regions (Gulf of Alaska and Canadian archipelago) closely modeled the
climate model data, whereas the precipitation pattern had a slight bias after the correction.





In order to incorporate glacier area changes to 2100, we follow the volume-area scaling method (Radić and Hock, 2011; Wahr et al., 2016). We use the initial conditions of glacier area and volume-area scaling (power law) relationship,

$$V \propto cS^{\gamma}, \tag{14}$$

to determine the glacier area and volume evolution over time (Bahr et al., 1997; Radić and Hock, 2010; Wahr et al., 2016).
Here, volume $V$ relates to $S$ the glacier area, where c is the constant in units of length raised to the power (3-2$\gamma$ ), and $\gamma$ is the dimensionaless scaling component based on theortical assumptions by Bahr et al. (1997). For glaciers in Gulf of Alaska, we consider $\gamma = 1.36$ and use $\gamma = 1.34$ for ice cap as in Arctic Canada North and South. The value of c = 0.03 to 0.026 under the density assumption of $\rho_{ice}$ (Bahr et al., 1997, 2015).

Following this, we use the relationship between volume and mass based on the density of ice ($\rho_{ice} = 917 \, \mathrm{kg \, m^{-3}}$) to calculate
volume rates as

$$V_{gl} = \frac{M_{gl}(t)}{\rho_{ice}}. \tag{15}$$

where $V_{gl}$ is the volume of glaciers and $M_{gl}(t)$ is the total mass balance of glaciers computed from equation 6. Here, our model does not incorporate the time series evolution of glacier area changes, since our model constrained by GRACE observations has secular and seasonal trends in mass balance. Therefore, we incorporate certain assumptions in the initial state of glacier
geometry. In order to test the effects of change in mass and volume loss based on steady state of glacier area and hypothetical conditions of mass loss, we perform a synthetic test and remove mass from glaciated area below $< 1500$ m or $< 600$ - 800 m (Gulf of Alaska and Canadian Archipelago) to understand the effects of glacier hypsometry.

In each model run for the period from 2002 to 2017, the model is tuned with three sensitivity parameters such that we obtain maximum variance between GRACE and modelled mass balance. For extrapolation of volume and mass loss rates, the main
three parameters are kept constant while tuning the other sensitivity parameters such as lapse rate, precipitation factor and precipitation lapse rate (Ref table 5). For initial conditions of volume, we considered $1.6 \times 10^4 \, \mathrm{km^3}$ for Gulf of Alaska North, $0.97 \times 10^4 \, \mathrm{km^3}$ for Arctic Canada South, and $3.2 \times 10^4 \, \mathrm{km^3}$ for Arctic Canada North ((Wahr et al., 2016)). The new rates of mass balance $\Delta M_{gl}$ are obtained for glaciers using the relationship between initial mass $M_0^{gl}$ and change in mass $\Delta M_{gl}(t)$ as,

$$\Delta M_{gl} = M_0^{gl} \left( \left[ 1 + 0.26 \frac{\Delta M_{gl}(t)}{M_0^{gl}} \right]^{3.78} - 1 \right). \tag{16}$$

The sea-level estimate (SLE) or change in sea-level is computed following

$$SLE = \frac{-1}{A_{ocean}} \sum_{i=1}^{n} M_i, \tag{17}$$

where we assume the area of ocean as $361 \times 10^6 \, \mathrm{km^2}$.



# 3 Results

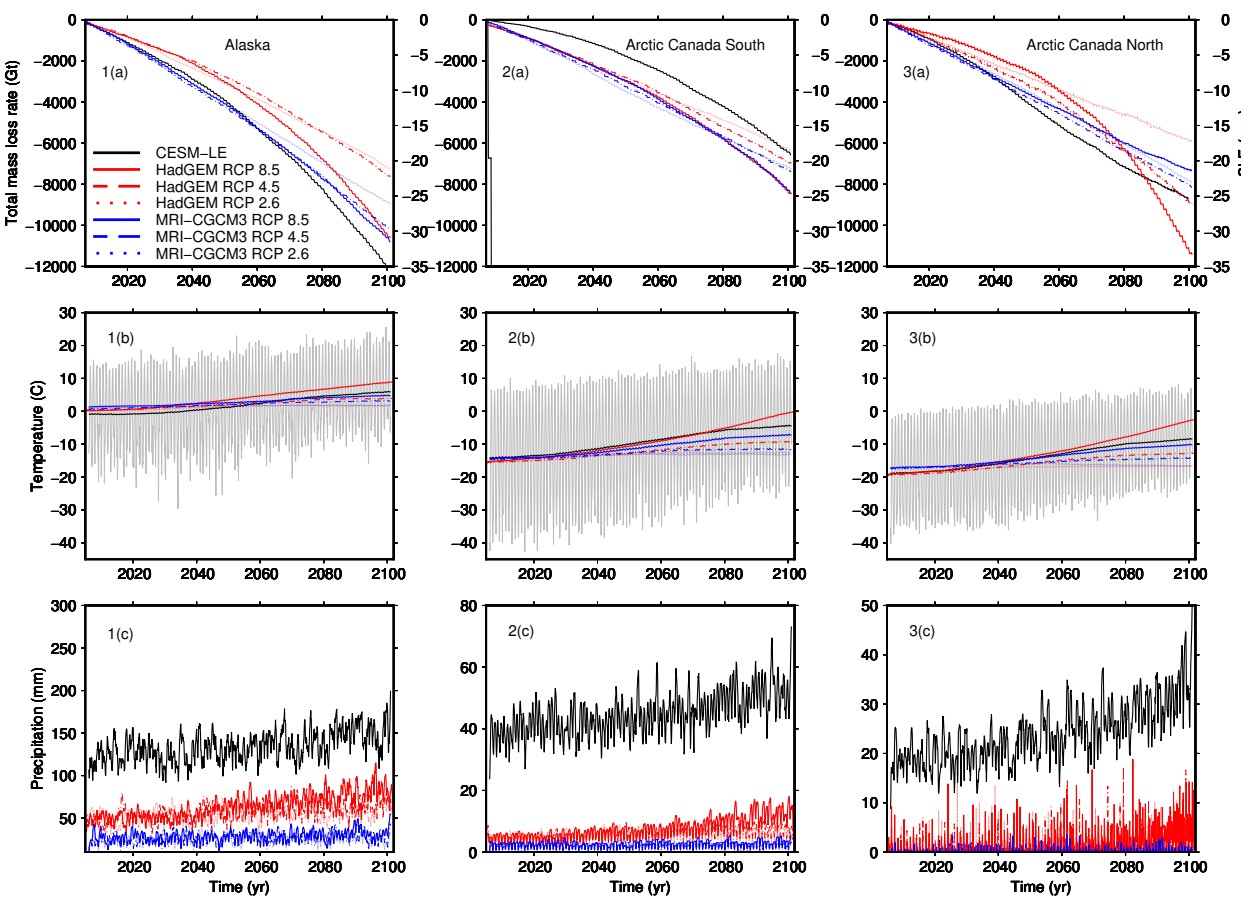

**Figure 2.** Projected future changes through 2100 in mass balance (top row), temperature (middle row), and precipitation (bottom row), for Alaska (first column), Arctic Canada South (second column), and Arctic Canada North (third column). The changes are shown for three different climate models (black, red, or blue), two of which are shown for several emission scenarios (solid, dashed, or dotted). Mean surface temperature and precipitation are obtained from grids within the glacier boundaries defined in Figure 1. Values from CESM-LE are indicated in black, values from HadGEM-ES in red, and MRI-CGCM3 in blue. Results for scenario RCP8.5 are shown with solid lines, RCP4.5 with dashed lines, and RCP2.6 with dotted lines.

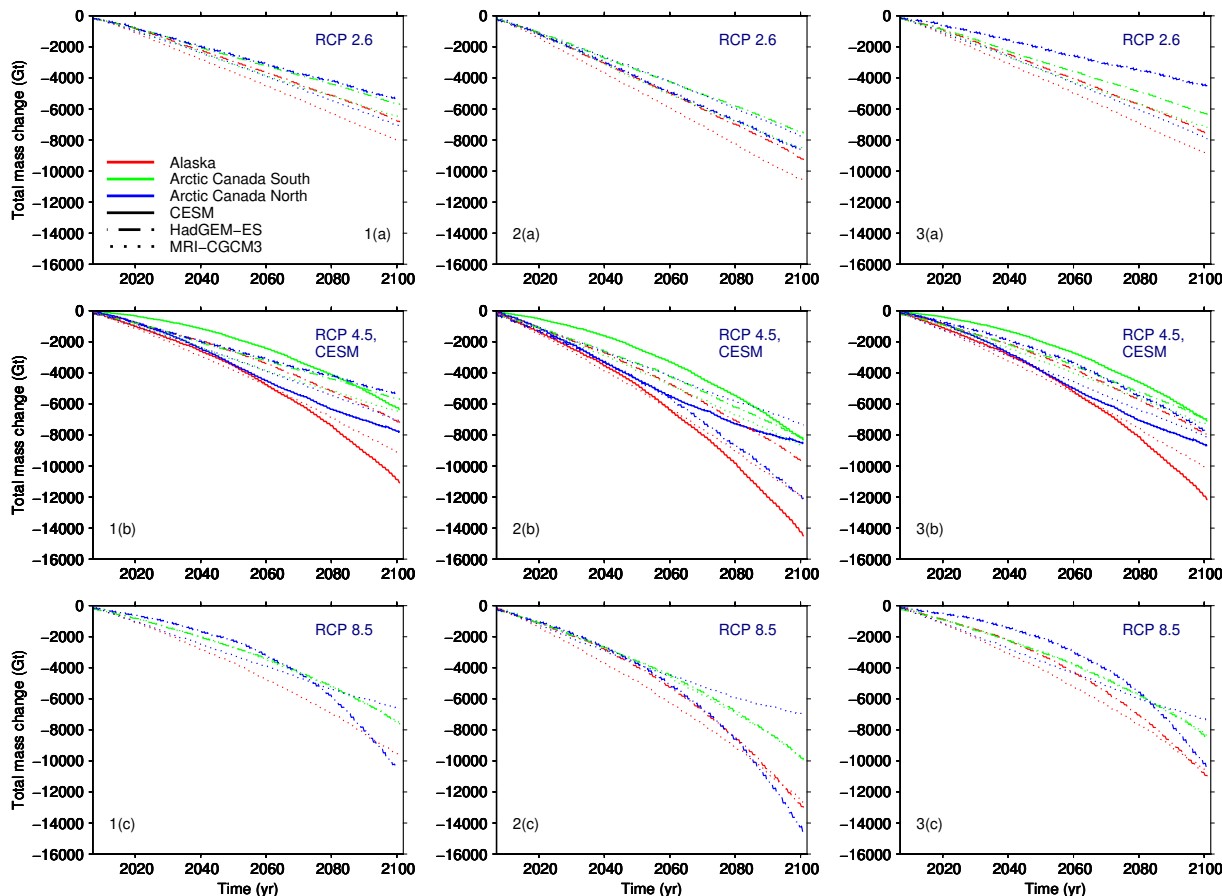

**Figure 3.** Projected future changes in mass balance through 2100 for different synthetic tests, shown for several climate models and emissions scenarios. The leftmost column 1 shows mass balance changes based on area changes with 10% removed from the terminus. The middle column 2 shows mass balance changes based on temperature increase of 1°K over standard. The right column 3 shows mass balance changes based on precipitation increase up to 10%. Line color indicates the region, while line type (solid or dashed) indicates the climate model. Climate scenarios include (a, top row) low emission RCP 2.6, (b, middle row) moderate emission RCP 4.5 and CESM, and (c, bottom row) high end emission scenario RCP 8.5.





We determine mass loss rates from GRACE and glacier model (respectively) that are $-36 \pm 4$ Gt yr$^{-1}$ and $-41 \pm 4$ Gt
yr$^{-1}$ for Gulf of Alaska, $-27 \pm 1$ Gt yr$^{-1}$ and $-43 \pm 2.7$ Gt yr$^{-1}$ for the Arctic Canada South, and $-38 \pm 1.3$ Gt yr$^{-1}$ and
$-45$ Gt yr$^{-1}$ $\pm 4$ Gt yr$^{-1}$ for the Arctic Canada North between 2002 and 2017 (Figure 1). By optimization of key parameters,
highlighted in Table 2, our model is able to explain about 87 to 88% of variance from GRACE observation, depending on
region (See figure A3).

The values from correlation and RMSE indicates that our model was able to explain the cumulative mass change obtained
from GRACE data. In Figure 1d–f we observe that our model does not explain much of the seasonality in Gulf of Alaska and
Arctic Canada North, whereas there is clear modeled seasonal signal from glaciers in the Arctic Canada South. By performing
model optimization in each region separately, the three main sensitivity parameters, $DDF, T_o$ and $K_o$ were invariably different
between the three regions (Table 5).

While we optimized our model against the regional mass balance, we also compared our model output against direct obser-
vations of mass balance from glaciers or ice caps in the Gulf of Alaska and Arctic Canada North (Figure A4). Gulkana and
Wolverine glacier in the Gulf of Alaska region have direct mass balance observations that overlap the GRACE observational
period (Figure A2). Gulkana glacier appears to be losing more mass than the Wolverine glacier, and our model explains 45%
of the variance with significant p-value. The Wolverine glacier model explained less variance with insignificant confidence
levels (Figure A5). In the Arctic Canada North, we found that direct observations from Meighen and White glacier explained
significant relationship with variance up to 84% and 45%, however, the observations from Devon ice cap was insignificant with
45% correlation (Figure A6). In the above comparison of direct observations, the three main sensitivity parameters were kept
constant as mentioned in Table 2 (highlighted). Optimization of direct observations was achieved mainly from tuning the lapse
rate in temperature and precipitation, which nearly represents the average elevation of glacier. This experiment indicated that
our regional model was able to represent local observations.

Future rates of mass loss varies from $-80$ Gt yr$^{-1}$ to $-112$ Gt yr$^{-1}$ in the Gulf of Alaska, $-75$ Gt yr$^{-1}$ and $-85$ Gt yr$^{-1}$
in Arctic Canada South islands, and $-64$ Gt yr$^{-1}$ and $-101$ Gt yr$^{-1}$ in Arctic Canada North under different climate emission
scenarios. Here, we considered the first ensemble, which is r1p1 in CMIP5 dataset or 001 experiment in CESM-LE climate
model data. Uncertainties in the volume and mass loss rates depends primarily on the (i) initial conditions of volume, (ii)
glacier hypsometry or area changes, and (iii) sensitivity to temperature and (iv) precipitation. For the volume loss rates, we use
the initial conditions to estimate the percentage of volume loss and its corresponding sea-level rise. Our model indicates that
the majority of volume loss occurs in Gulf of Alaska and Arctic Canada South, which corresponds to 17–20% volume loss or
a sea level equivalent of 27–37 mm for Gulf of Alaska and 18–25 mm for Arctic Canada South Island. Arctic Canada North
contributes to a volume loss 9–12% that corresponds to a sea level equivalent of 22–25 mm.



# 4 Discussion

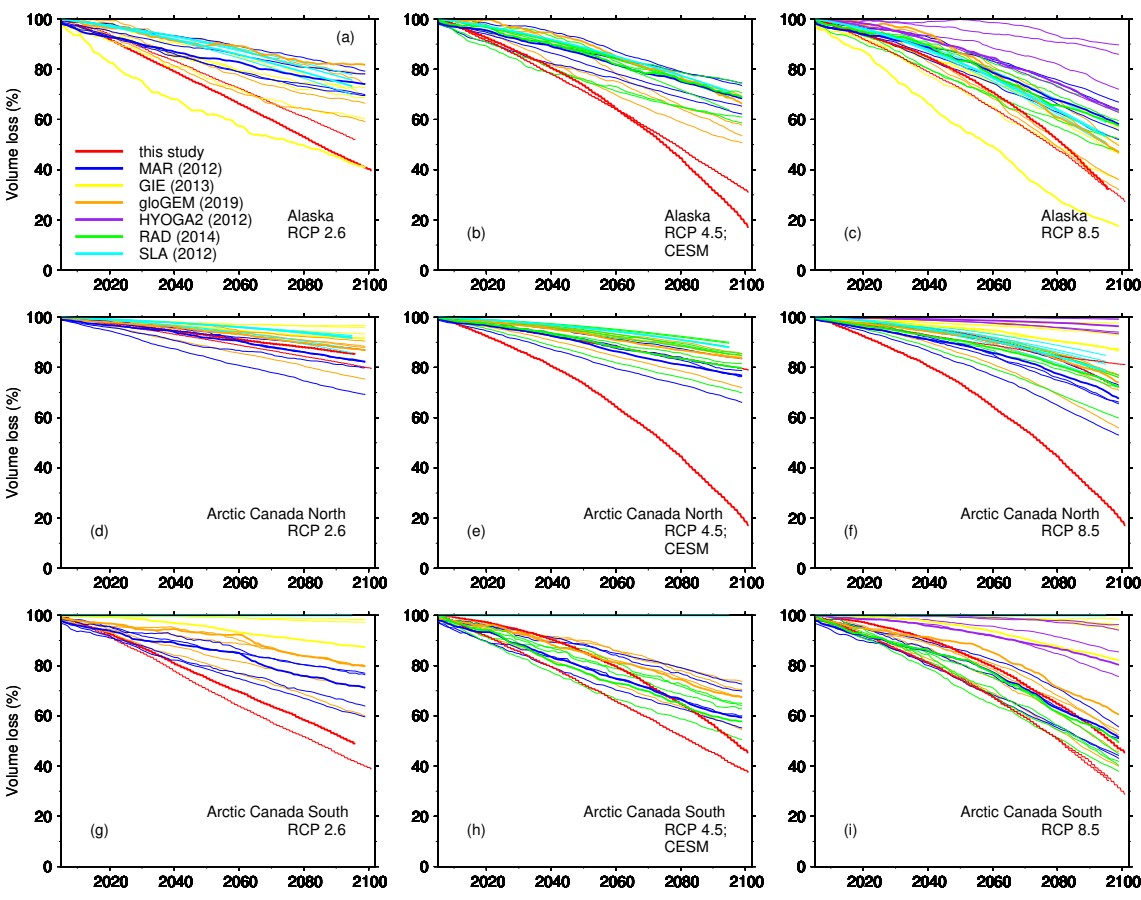

**Figure 4.** Comparison of sea-level estimates from the existing global glacier models for Gulf of Alaska, Arctic Canada North and South for the period from 2006 to 2100. The rates of SLE are obtained from Hock et al. (2019) supplementary table. We use a similar notation to refer the glacier models (Last name and year of the publication). Global glacier models include results from Marzeion et al. (2012), Slangen et al. (2012), Giesen and Oerlemans (2013), Radić et al. (2014) and Hock et al. (2019).

Our observations of volume and mass loss rates derived from the optimization of GRACE dataset are consistent with several recent studies and in accordance with the IPCC (2019). We adopted a unique approach in modeling mass balance rates compared to the existing methods that are highly dependent on direct observations for extrapolating regional and global mass balance. We discuss the results from the perspectives of (i) model comparison with previous studies, (ii) model performance and sensitivity analysis, (iii) model representation with the local observations, and (iv) uncertainties in projected estimates of SLE and error propagation.





## 4.1 Model comparison with previous studies

Even though the modeling framework adopted in our study is not directly comparable to the existing glacier models, the results are in accordance with the published sea-level estimates in the IPCC special report (IPCC, 2019). In the glacier model intercomparsion project (GlacierMIP), six glacier models have been compared in terms of their modeling capabilities and
uncertainties in the future projection. Like the existing glacier models, we used a degree day method to model the mass balance from gridded temperature and precipitation. In the previous studies, either individual glaciers from the RGI inventory or the total glacier area grouped in elevation bins were calibrated with WGMS direct observations for all regions together (Marzeion et al., 2012; Giesen and Oerlemans, 2013; Hirabayashi et al., 2013; Radić et al., 2014). Based on the model calibration with direct measurements, the volume and mass loss rates were extrapolated for future projections. One of the drawback from this
approach was large uncertainties in the regions of sparse direct observations or high topographic regions such as the High Mountain Asia or Andes. Some of the results from these models could not be compared directly due to the differences in glacier inventory version and meteorological inputs from either ERA interim or CRU datasets.

This problem was circumvented by the use of standardized regional mass balance by Gardner et al. (2013) to calibrate the model (Huss and Hock, 2015). In this way, every glacier in the RGI inventory is tuned to match the regional mass balance
between 2003 and 2009. This method does not attempt to model the local observations from individual glaciers but does solve the uncertainties from regions deficient in direct observations. Model calibration also included glacier thickness measurements for all glaciers to incorporate dynamics in the future projections of volume and mass loss, instead of volume area scaling. Huss and Hock (2015) also included calibration for grounded ice that is already displaced into ocean from calving fronts. It was estimated that error from displacements due to calving fronts contributed about 10-14% to the global sea-level.

One of the advantages from our model is that it does not require calibration from direct mass balance observations, which are spatially and temporally sparse, with annual field observations. Instead, we calibrate our model using GRACE observations with higher spatial and temporal resolution, which are known for estimating regional mass balance very accurately (e.g. Harig and Simons, 2012). The regionally calibrated parameters ($DDF$, $T_o$ and $K_o$) are unique for Alaska, Arctic Canada North and South, and they are capable of representing direct observations from individual glaciers (Figure A4). In this way, we address
the bias due to sampling issue from direct observations.

In the existing models, calibration did not account for dynamics from glaciers which could reflect the response time of glacier leading to cumulative bias for longer timescales. Only the model by Marzeion et al. (2012) included relaxation time in glaciers for the present and future rates of volume and mass loss. Our model constrained by GRACE intrinsically accounts for higher order dynamics, evident from the seasonal signal at monthly intervals (Fig 1). It is also one of the reason that we obtain
good confidence in the mass balance time series from individual glaciers such as the Gulkana, Wolverine, Devon, White and Meighen icecap (Figure A5 and A6).




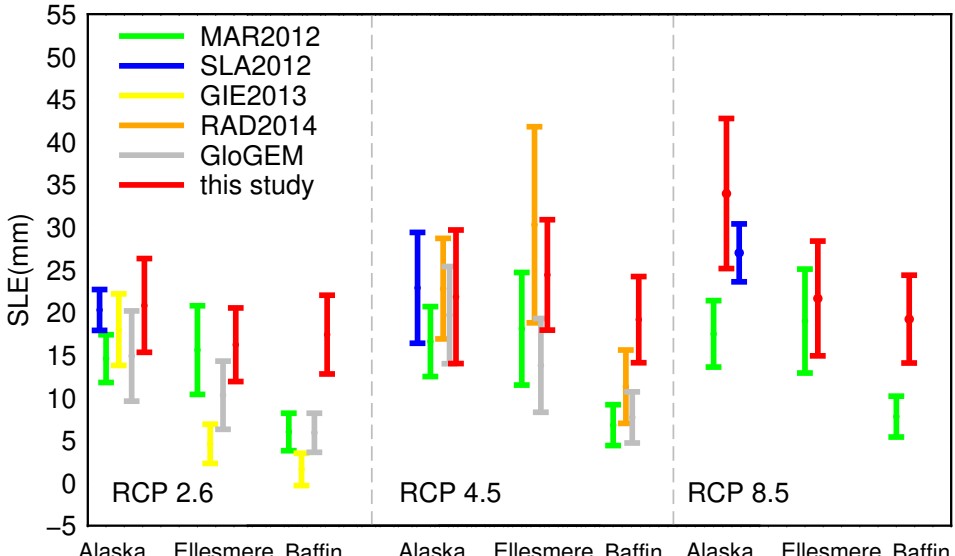

**Figure 5.** Comparison of sea-level estimates from the existing global glacier models for Gulf of Alaska, Arctic Canada North and South for the period from 2006 to 2100. The rates of SLE are obtained from Hock et al. (2019) supplementary table. We use a similar notation to refer the glacier models (Last name and year of publication). Global glacier models include results from Marzeion et al. (2012), Slangen et al. (2012), Giesen and Oerlemans (2013), Radić et al. (2014) and Hock et al. (2019).

From figure 3, we find that the projected mass changes in the Arctic Canada South (Baffin) region are nearly similar compared to the Arctic Canada North. Our model projects a volume loss of 16–19% for Gulf of Alaska, 18–20% for Arctic Canada South, and 9–12% for Arctic Canada North (Table 3). Our observations are somewhat consistent with Huss and Hock (2015)
for Arctic Canada South, where the volume loss is -37 ± 14 % for RCP 4.5, while our observations indicated a volume loss of ~20%. We do not attempt to compare the initial state of glacier thickness from other studies due to different versions of RGI inventory and assumptions in volume-area scaling.

We tested the sensitivity of our model with three experiments that influences the mass and volume rates. In the dynamic adjustments of glaciers for mass loss, we assume that the glacier will lose mass near the terminus, therefore we remove 10%
of mass from the terminus region (Figure 3). We also tested with a temperature increase of 1°K and precipitation change of +10%. Our model is highly sensitive to temperature changes with mass loss and SLE increase by 11–28 Gt $yr^{-1}$ and 6–11 mm, compared to the normal conditions (Table A2). Comparatively, our model was less sensitive to changes in precipitation.

We compare our results to the projection estimates from Glacier model intercomparsion (GlacierMIP) project that incorporates six published global glacier models (Hock et al., 2019) (Figure 5). Some of the previous models did not attempt mass
balance projection for RCP 8.5 or RCP 2.6, therefore we use RCP 4.5 as an intramodel comparsion and later, we compare with respect to specific regions. From Figure 5, we observe consistent results for Gulf of Alaska with other models, with SLE at 21.85 mm (current model), 22.8 mm (RAD2014) and 16.6 mm (MAR2012). The level of uncertainties (error bars) match with both the models. For the Arctic Canada North at RCP4.5, we fall within the error bars of RAD2014, and mass loss rates are slightly higher from our model compared to GloGEM and MAR2012. Most previous models estimated significantly lower



rates of SLE for Arctic Canada South than the North (e.g. Marzeion et al. (2012) estimated 19.0±6.1 for Arctic Canada North and 7.8 ± 2.4 for Arctic Canada South islands at RCP 4.5). Our model, in contrast exhibited higher SLE from ACS that is on par with our estimate of SLE from Arctic Canada North (21.45 mm vs. 25.23 mm). Additionally, the Gulf of Alaska region displays the highest sensitivity to the specific emissions scenario used (RCP 2.6/4.5/8.5) with SLE increasing more in scenarios with higher temperatures. Overall, we find better agreements for Gulf of Alaska for all RCP scenarios, slight deviation in the Arctic Canada North and much higher SLE rates for Arctic Canada South (Figure 5).

We believe that the high rates of projected mass loss from Arctic Canada South are due to the model calibration from the GRACE dataset. Our observations from GRACE indicate that glaciers in Arctic Canada South have high rates of mass loss, despite the smaller proportion of glaciers (Figure 1). Our present rates of mass loss from GRACE are nearly consistent with results by Wouters et al. (2019) for Arctic Canada North and Arctic Canada South, which had mass change of $-35.8 \pm 3.5$ $Gtyr^{-1}$ and $-32.5 \pm 7.8$ $Gtyr^{-1}$ for the period between April 2002 and April 2016. Similarly, modeled mass balance using RACMO2.3 indicated comparable mass loss rates for Arctic Canada North and Arctic Canada South ($-24.7$ Gt yr$^{-1}$ vs. $-21.9$ Gt yr$^{-1}$) from 1996 to 2015 (Noël et al., 2018). This indicates that the mass balance response has increased over the last decade in both Arctic Canada North and South in contrast to the inference from glacier models that reported smaller estimates of SLE for Arctic Canada South (Hock et al., 2019). We think that global models initialized from direct observations and inventories does not capture the sensitivity from small glaciers and ice caps, as in Arctic Canada South. There is an uncertainty of 18–30%, specifically due to sensitivity from small glaciers (Huss and Hock, 2015; Hock et al., 2019). This leads to the erroneous conclusion that large glacierized areas in the Arctic tend to lose more mass than small glaciers in the Arctic Canada South.

## 4.2 Model performance and sensitivity analysis

We consider a different approach to determine the glacier volume and mass loss rates incorporating higher order of glacier dynamics without the need for inputs from direct observations or glacier thickness measurements (Radić et al., 2014; Huss and Hock, 2015). In the Figure 1d and f, the modelled rates of mass balance agree well for Gulf of Alaska and Arctic Canada North, however there is a discrepancy in the mass loss rates for Arctic Canada South. The modelling parameters indicated in Table 5 achieved good correlation and model convergence for all three regions, however there is slightly a large discrepancy in mass loss rates between GRACE and model for Arctic Canada South. We could have acheived lower discrepancy between the GRACE and modelled time series for ACS, but this will involve compromising some of the model parameters for convergence. This could be unrealistic, for example, our model cannot have precipitation lapse rate > 1500 m as it does not represent the ELA of glaciers in ACS. Also, we found that using these unrealistic values in the model parameters significantly affected in the future projections of mass loss rates and SLE. We agree that our estimates of GRACE mass loss rates were slightly lower at -27 Gtyr$^{-1}$, compared to recent estimate of -32 ± 8 Gtyr$^{-1}$ (Wouters et al., 2019). We also note that the seasonality and inter-annual variability present in GRACE data is not well represented in our model mass balance time series (Figures 1 d–f). GRACE is well known for capturing the seasonality in mass balance, whereas the modeled mass loss rates from reanalysis products often have large biases occurring from poor representation of precipitation in cold regions, either at high latitude or high elevation (Behrangi et al., 2016).





Previously, GRACE observations has been used in modeling sub-annual mass balance from glaciers in the Gulf of Alaska
with high degree of fidelity in modeling local observations (Arendt et al., 2008, 2013; Luthcke et al., 2013). Monthly GRACE
observations track the seasonality from glaciers, up to a variance of 75% and 56% from summer and winter balances respec-
tively (Arendt et al., 2013). We use this rationale to model the mass balance from individual glaciers, constrained by GRACE
observations.

While we employed seven sensitivity parameters to model the volume and mass loss rates from three regions (See Table
5) we optimize our model based on three main parameters ($DDF$, $T_o$, and $K_o$) to constrain our model against GRACE mass
balance timeseries (Wahr et al., 2016). This method eliminates the need for extrapolation of direct observations for regional
mass balance and SLE as in Radić et al. (2014). Further, we find that the degree day factor ($DDF$) and threshold temperature
($T_o$) varies between the three region. The $DDF$ for glaciers in Gulf of Alaska is 2.3 mm $d^{-1}$ $C^{-1}$, whereas for Arctic Canada
North and South, the DDF is slightly higher with values of 3 mm $d^{-1}$ $C^{-1}$ and 4 mm $d^{-1}$ $C^{-1}$, respectively. This is due
to higher mass balance sensitivity of glacier with temperature in the Arctic regions (Hock, 2003; De Woul and Hock, 2005;
Braithwaite and Raper, 2007). Similarly, the threshold temperature $T_o$ varies depending on the average temperature of the
region, which is larger for Arctic regions compared to the Gulf of Alaska. By optimizing these three parameters, our modeled
mass balance was able to explain $\sim 90\%$ of the variance from GRACE observations.

Similar to Radić et al. (2014), we found that our model is sensitive to environmental lapse rate ($lp$) and temperature lapse
rate ($\Delta h$). The values for environmental lapse are based on Gardner et al. (2009) and Braithwaite (2008). We introduced
uncertainties at $1\sigma$ in the DDF, $T_o$ and $K_o$ to obtain the bias in modeled mass balance. We found that the error was $-4$ Gt or
1.1 m.w.e for Gulf of Alaska, $-2.7$ Gt or 0.88 m.w.e for Arctic Canada South and $-4.0$ Gt or 1.0 m.w.e for Arctic Canada North
between 2002 and 2017. This error is slightly higher than bias obtained from Radić et al. (2014) and Marzeion et al. (2012)
who's error estimates were $-0.30$ and $-0.44$ m.w.e for Gulf of Alaska, $-0.12$ and $-0.18$ for Arctic Canada North (Ellesmere)
and $-0.24$ and $-0.19$ for Arctic Canada South (Baffin). Their smaller value in bias is due to extrapolation of regional mass
balance from a sample of $\sim 20$–$34$ glaciers from these regions and a different version of RGI inventory. In contrast, GRACE
regional mass balance includes observations from all glaciers with proper representation of summer and winter balance without
the need for scaling from direct observations.

### 4.3   Comparison with local observations

In the Gulf of Alaska, Gulkana and Wolverine glacier have direct mass balance data that overlap with our observational period
from 2002 to 2017. In order to model the local observations, the three main sensitivity parameters were kept constant, while
adjusting the other parameters until the model bias is minimized (Table 5). Table A3 provides the list of sensitivity values
used in modeling in-situ observations from the GRACE constrained model. We found a good correlation of 0.45 (p<0.09) for
Gulkana glacier, whereas modeled mass balance from Wolverine glacier had a correlation of 0.24 (p<0.38) with insignificant
p-value (Figure A5). GRACE observations were able to represent in-situ mass balance from Gulkana and Wolverine glacier
for highly negative mass balance during 2004 and 2013 (Luthcke et al., 2008; Wouters et al., 2019). These studies, like our
model indicated that the temperature is a contributing factor for glacier melt in the Gulkana glacier. Our model did not exhibit





good correlation for Wolverine glacier due to its proximity near the coast, with mass loss rates controlled by precipitation. The result was similar to the approach where individual mascons from GRACE observation were compared with direct observations

(Arendt et al., 2013).

In the Arctic Canada North, our model was able to represent in-situ observations from Devon ice cap (r = 0.42, p = 0.13), Meighen ice cap (r = 0.84, p = 0.001) and White glacier (r = 0.44, p = 0.09) (Figure A6). Most of the glaciers and ice caps are known to be losing mass rapidly in the 21st century based on a series of in-situ and satellite observations (Bezeau et al., 2013; Noël et al., 2018; Cook et al., 2019). Agreement of modelled mass balance is consistent for Meighen glacier, with

slight inconsistency for Devon ice cap owing to different input datasets and modeling techniques (Lenaerts et al., 2013). These observations indicate that our model constrained by GRACE is capable of representing in-situ mass balance.

### 4.4    Uncertainties from climate model and GRACE

For our projections, we used the recent CESM-LE, which is one of the latest generations of Earth Systems Models and a group member of Coupled Model Intercomparison Project Phase 5. Although we could not compare our results of CESM-LE with

the other 11 models used in previous studies, we found a good agreement for Alaska using HadGEM-ES with the Huss and Hock (2015) model (Supplementary table 9, page 15). This confirms that our GRACE constrained model is able to replicate a model that accounted for glacier thickness change and frontal ablation. Here we do not aim to understand the uncertainties from different climate inputs such as the ERAI, CMIP5 or CESM-LE. Therefore, we have included $1\sigma$ error in the mass loss time series to incorporate the various input errors. Our model is able to capture $\sim 90\%$ of the variance from GRACE observations,

which indicates that there is a certain loss of information from the model itself. This includes an uncertainties from the three sensitivity parameters such as the $DDF$, $T_o$ and $K_o$ and the GRACE solutions.

### 5    Conclusions

We present estimates of glacier volume and mass loss from a GRACE constrained glacier model that can be compared with the recent glacier model intercomparison project (GlacierMIP). We have three key findings from our glacier model and they

are as follows. (a) We have demonstrated that the regional bias can be significantly low compared to calibration from direct observations, during the observational period. This is due to the use of GRACE solutions for regional mass balance. This assumption holds true for any RGI glacier region, especially with HMA, Andes or Caucasus with complex topography. (b) Our model is able to represent local observations with good level of confidence in the mass balance timeseries. The unique sensitivities values obtained for each region represents the mass balance change from individual glaciers. This is useful for

understanding the response of benchmark glaciers under the present and future conditions of warming without the need for conventional field observations.

(c) We find projected mass losses from the Arctic Canada South that are far higher than other studies, and which are on par with losses from the Arctic Canada North. This indicates that Arctic Canada South has greater sensitivity in the recent decade, and our model is able to capture this sensitivity. In order maintain the consistency with global glacier models, we have used



the same sets of input climate datasets, glacier inventories, and a similar approach in the degree-day model. We find that our model is able to represent higher order of glacier responses under the transient conditions of temperature and precipitation.

Future work will involve understanding the response of small glaciers to climate change. With increase in temperature, small glaciers respond to climate change with greater volume loss. We plan to test our model from a global perspective, in accordance with the GlacierMIP.

*Acknowledgements.* This work was supported by the TRIFF-WEES Program at the University of Arizona. Figures were plotted using the Generic Mapping Tools (Wessel et al., 2013).



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





**Table 1.** Conditions used in the sensitivity analysis for the optimization of GRACE and ERA-based mass balance

| Sensitivity parameter | Description | Starting values in region | | | Units |
| --- | --- | --- | --- | --- | --- |
| | | Gulf of Alaska | Arctic Canada South | Arctic Canada North | |
| $\mathbf{T_o}$ | ERA threshold temperature | 274.15 | 261.15 | 272.15 | K |
| **DDF** | Degree day factor | 2.3 | 4 | 3 | $\frac{mm}{d\,^{\circ}C}$ |
| $\mathbf{K_o}$ | Precipitation correction factor | 0.69 | 0.63 | 0.69 | – |
| $\Delta h$ | Temperature lapse rate | 1700 | 300 | 900 | m |
| $d_{prec}$ | Precipitation gradient | 0.13 | 0.13 | 0.13 | – |
| $\Delta p$ | Precipitation lapse rate | 1000 | 900 | 1200 | m |
| lp | Environmental lapse rate | 0.0065 | 0.0045 | 0.0062 | $\frac{m}{^{\circ}C}$ |

**Table 2.** Details of climate models used in the estimation of future mass loss and sea-level rates

| Model | Climate emission scenario | Center and location | Spatial resolution | Reference |
| --- | --- | --- | --- | --- |
| CESM-LE | Similar to RCP 8.5 | University of Colorado, Boulder | $1^{\circ} \times 1^{\circ}$ | Kay et al. (2015) |
| HadGEM-ES | RCP 2.6, 4.5, 8.5 | Met Office Hadley, UK | $1.88^{\circ} \times 1.25^{\circ}$ | Jones et al. (2011) |
| MRI-CGCM3 | RCP 2.6, 4.5, 8.5 | Meterological Research Institute, Japan | $1.12^{\circ} \times 1.12^{\circ}$ | Yukimoto et al. (2012) |



**Table 3.** Estimates of volume and mass loss rates from Gulf of Alaska, Arctic Canada South and Arctic Canada North, for the period between 2006 and 2100.

| Region | Model | Initial glacier volume (km$^3$) | Initial glacier area (km$^2$) | $\Delta V$ (%) | SLE (mm) | MB rate (Gtyr$^{-1}$) |
|---|---|---|---|---|---|---|
| Gulf of Alaska | CESM | 1.6 x 10$^4$ | 5.7 x 10$^4$ | 19 | 37.33 | -109.66 |
| | HadGEM RCP 8.5 | | | 17 | 30.83 | -106.09 |
| | HadGEM RCP 4.5 | | | 16 | 22.77 | -81.75 |
| | HadGEM RCP 2.6 | | | 17 | 21.82 | -79.84 |
| | MRI RCP 8.5 | | | 19 | 33.11 | -112.49 |
| | MRI RCP 4.5 | | | 19 | 31.36 | -107.96 |
| | MRI RCP 2.6 | | | 19 | 27.45 | -95.65 |
| Arctic Canada South | CESM | 0.97 x 10$^4$ | 3.87 x 10$^4$ | 13 | 12.9 | -57.18 |
| | HadGEM RCP 8.5 | | | 19 | 23.88 | -83.43 |
| | HadGEM RCP 4.5 | | | 18 | 20.26 | -72.34 |
| | HadGEM RCP 2.6 | | | 19 | 18.53 | -66.93 |
| | MRI RCP 8.5 | | | 19 | 25.85 | -85.92 |
| | MRI RCP 4.5 | | | 20 | 22.64 | -78.46 |
| | MRI RCP 2.6 | | | 20 | 22.19 | -77.75 |
| Arctic Canada North | CESM | 3.2 x 10$^4$ | 1.05 x 10$^5$ | 10 | 21.71 | -101.15 |
| | HadGEM RCP 8.5 | | | 10 | 31.11 | -100.86 |
| | HadGEM RCP 4.5 | | | 11 | 25.49 | -90.81 |
| | HadGEM RCP 2.6 | | | 9 | 17.36 | -63.13 |
| | MRI RCP 8.5 | | | 11 | 22 | -78.89 |
| | MRI RCP 4.5 | | | 12 | 24.97 | -85.48 |
| | MRI RCP 2.6 | | | 11 | 24.26 | -83.38 |





**Appendix A:**

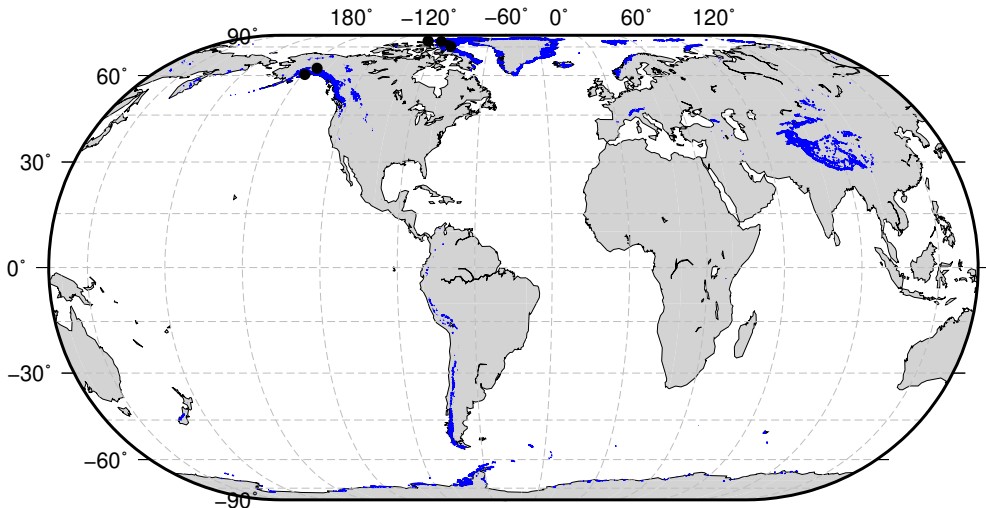

**Figure A1.** Location of glaciers from Gulf of Alaska and Arctic Canada North with direct mass balance observations for the period 2002 to 2017 (indicated in black). These five glaciers are modelled for local mass balance using our model, and this is indicated on a global map with the RGI glacier outlines (blue)

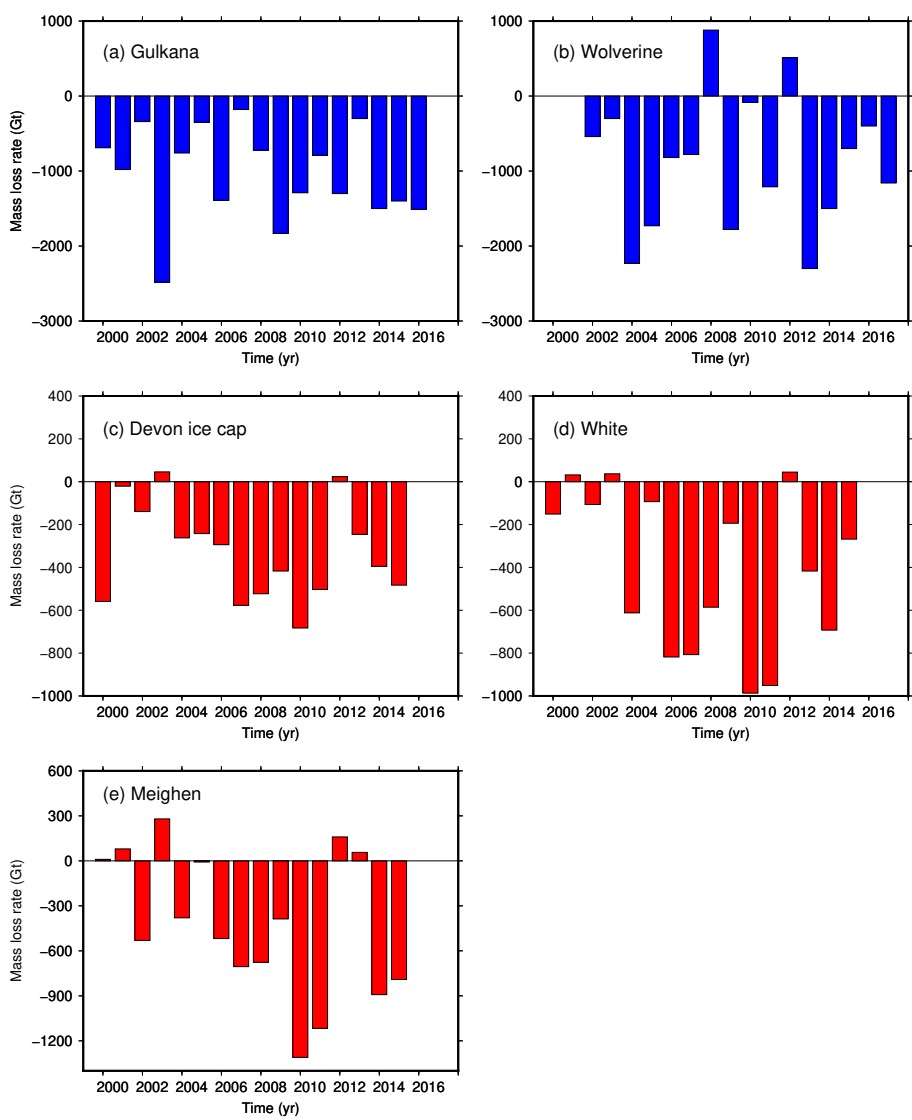

**Figure A2.** Direct observations of mass balance from WGMS for glaciers in the Gulf of Alaska (a) Gulkana, (b) Wolverine, and for glaciers and ice caps in the Arctic Canada North that includes (c) Devon ice cap, (d) White glacier and (e) Meighen ice cap. Note that the WGMS observations are provided annually for each hydrological year.


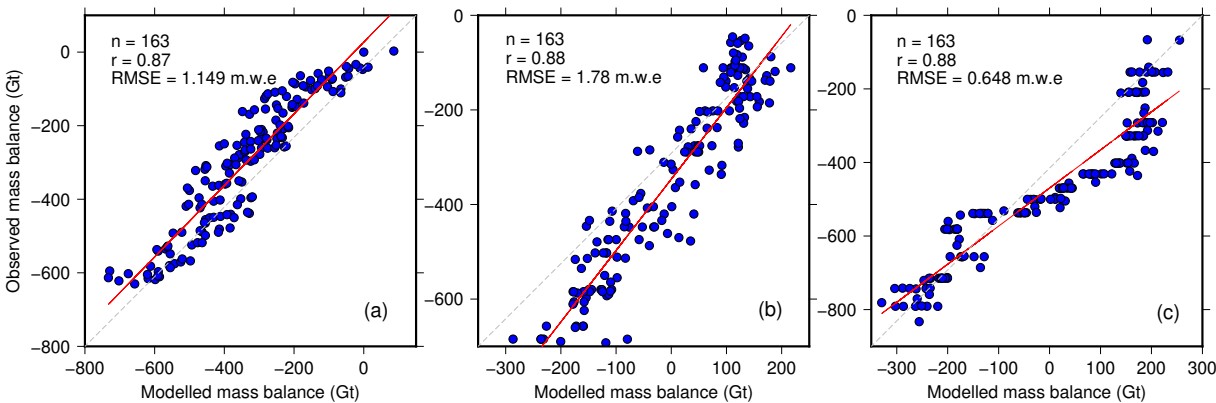

**Figure A3.** Bias in the mass loss rates obtained from GRACE observations (observed) and the ERA based mass balance (modelled), indicated as Gtyr$^{-1}$ in Figure 1 for three regions: (a) Gulf of Alaska, (b) Arctic Canada South (Baffin), and (c) Arctic Canada North (Ellesmere). RMSE and correlation was obtained from optimisation of GRACE observations with modelled mass balance.

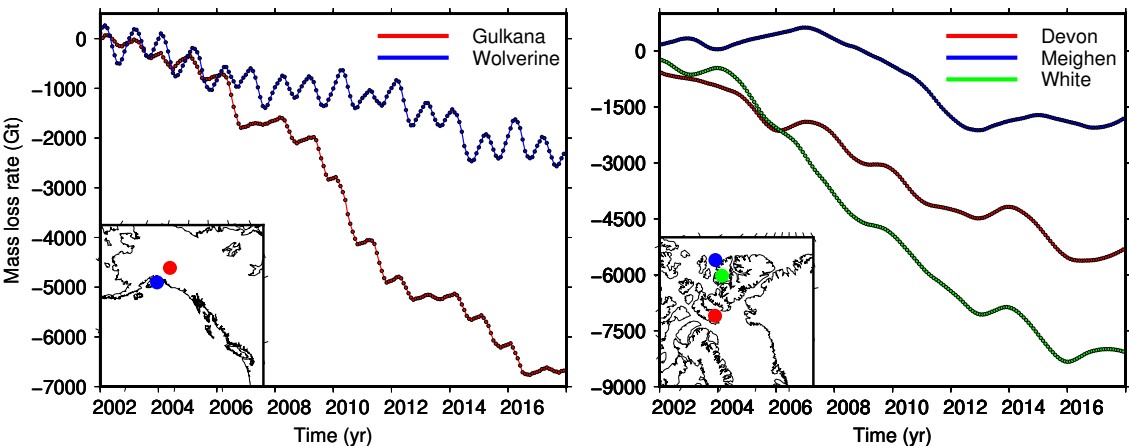

**Figure A4.** Modelled observations of direct mass balance based on the optimisation of GRACE and ERA-Interim dataset (a) Modelled mass balance for glaciers in the Gulf of Alaska, (b) Modelled mass balance for glaciers and ice cap in the Arctic Canada North for the period from 2002 to 2017.

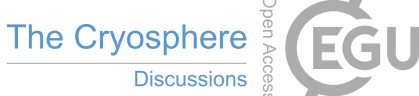

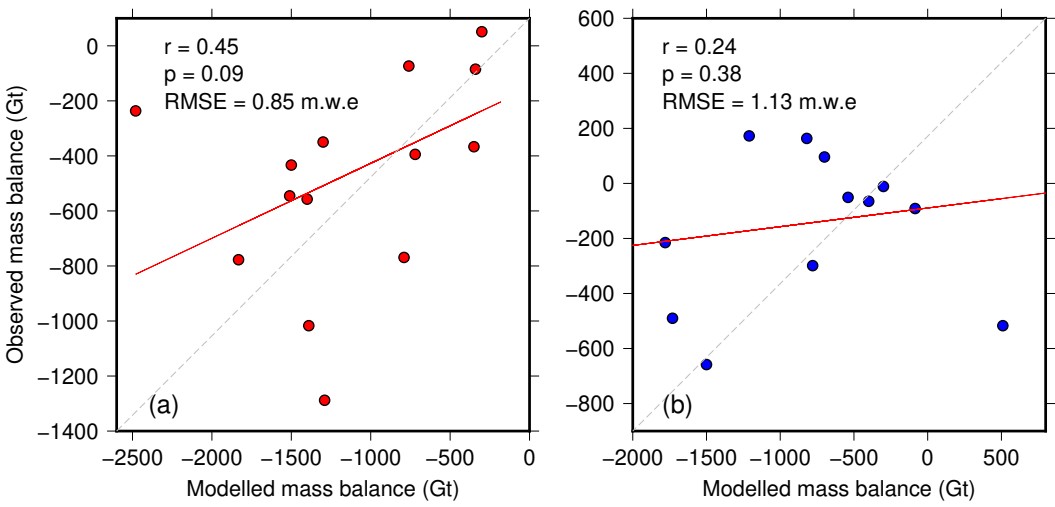

**Figure A5.** Modelled observations of direct mass balance based on the optimisation of GRACE and ERA-Interim dataset (a) Modelled mass balance for glaciers in the Gulf of Alaska, (b) Modelled mass balance for glaciers in Arctic Canada North for the period from 2002 to 2017.

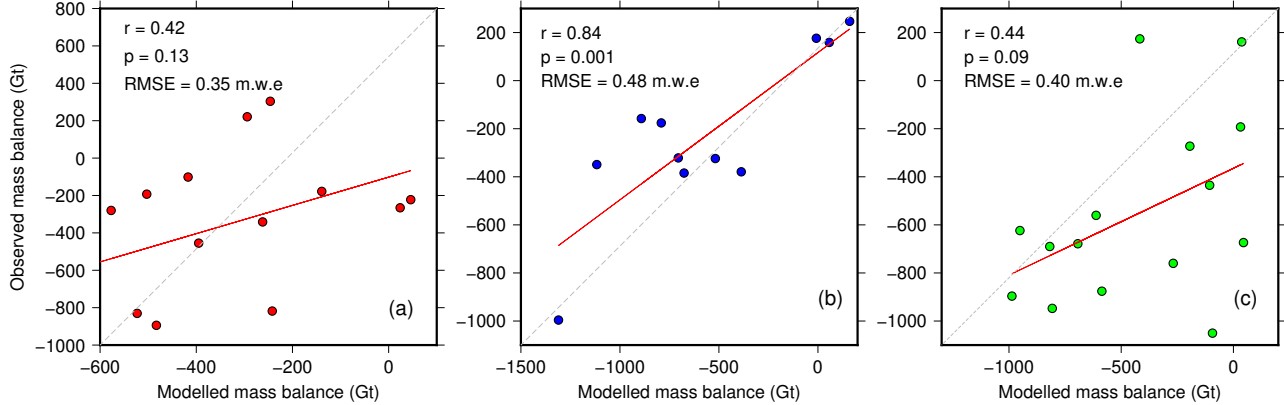

**Figure A6.** Modelled observations of direct mass balance based on the optimisation of GRACE and ERA-Interim dataset (a) Modelled mass balance for glaciers in the Gulf of Alaska, (b) Modelled mass balance for glaciers in Arctic Canada North for the period from 2002 to 2017.





**Table A1.** Mass loss and sea-level rates from Gulf of Alaska North, Arctic Canada South and Arctic Canada North from 2006 to 2100, under an area loss from the terminus

| Region | Model | $\Delta$T ($^{\circ}$C) | $\Delta$P (mm) | $\Delta$V (%) | SLE (mm) | MB rate (Gtyr$^{-1}$) |
|--------|-------|------|------|------|----------|-----------|
| Gulf of Alaska | CESM-LE | -1.65 | 14.24 | 18.5 | -33.96 $\pm$ 8.72 | -113.73 $\pm$ 15.58 |
| | HadGEM RCP 8.5 | -6.82 | 7.9 | 16.6 | -29.92 $\pm$ 7.83 | -99.04 $\pm$ 12.82 |
| | HadGEM RCP 4.5 | -8.48 | 7.33 | 15.8 | -21.85 $\pm$ 5.79 | -74.34 $\pm$ 9.38 |
| | HadGEM RCP 2.6 | -8.84 | 6.99 | 16.3 | -20.83 $\pm$ 5.50 | -71.42 $\pm$ 8.90 |
| | MRI RCP 8.5 | -5.92 | 0.68 | 18.3 | -29.8 $\pm$ 8.55 | -101.25 $\pm$ 12.10 |
| | MRI RCP 4.5 | -6.27 | 0.64 | 19 | -28.23 $\pm$ 8.19 | -97.16 $\pm$ 11.44 |
| | MRI RCP 2.6 | -7.24 | 0.62 | 18.6 | -24.71 $\pm$ 7.25 | -86.09 $\pm$ 10.36 |
| Arctic Canada South | CESM-LE | -10.36 | 6.07 | 16.4 | -19.23 $\pm$ 5.93 | -67.4 $\pm$ 12.15 |
| | HadGEM RCP 8.5 | -12.84 | 1.7 | 18.6 | -23.22 $\pm$ 5.15 | -75.09 $\pm$ 12.19 |
| | HadGEM RCP 4.5 | -15.25 | 1.7 | 18.2 | -19.17 $\pm$ 5.05 | -65.1 $\pm$ 10.67 |
| | HadGEM RCP 2.6 | -15.98 | 1.9 | 18.2 | -17.41 $\pm$ 4.62 | -60.24 $\pm$ 9.68 |
| | MRI RCP 8.5 | -11.57 | 1.17 | 16.7 | -20.96 $\pm$ 5.90 | -77.32 $\pm$ 13.41 |
| | MRI RCP 4.5 | -13.12 | 1.19 | 17 | -18.34 $\pm$ 5.35 | -70.62 $\pm$ 12.37 |
| | MRI RCP 2.6 | -13.55 | 1.14 | 17.3 | -17.98 $\pm$ 5.31 | -69.97 $\pm$ 12.02 |
| Arctic Canada North | CESM-LE | -18.76 | 2.67 | 11.7 | -21.71 $\pm$ 6.73 | -89.62 $\pm$ 14.21 |
| | HadGEM RCP 8.5 | -8.08 | 16.59 | 10 | -30.98 $\pm$ 7.84 | -96.56 $\pm$ 23.06 |
| | HadGEM RCP 4.5 | -10.08 | 18.34 | 10 | -24.42 $\pm$ 6.48 | -82.92 $\pm$ 21.00 |
| | HadGEM RCP 2.6 | -11.84 | 16.31 | 9 | -16.23 $\pm$ 4.32 | -55.84 $\pm$ 18.70 |
| | MRI RCP 8.5 | -19.2 | 8.02 | 11 | -18.23 $\pm$ 5.38 | -70.98 $\pm$ 10.62 |
| | MRI RCP 4.5 | -20.12 | 5.49 | 11.6 | -20.24 $\pm$ 5.83 | -76.93 $\pm$ 10.98 |
| | MRI RCP 2.6 | -21.34 | 4.85 | 11 | -19.66 $\pm$ 5.69 | -75.04 $\pm$ 10.79 |





**Table A2.** Mass loss and sea-level estimates from three regions based on the conditions in temperature (+1K) and 10% increase in precipitation, from 2006 to 2100

| Region | Model | Org T | ΔT + 1K | SLE ± 1σ (mm) | MB rate ± 1σ (Gtyr⁻¹) | Org P | ΔP +10 % | SLE ± 1σ (mm) | MB rate ± 1σ (Gtyr⁻¹) |
|---|---|---|---|---|---|---|---|---|---|
| Gulf of Alaska | CESM-LE | -1.65 | -0.65 | -44.46 ± 11.43 | -149.29 ± 16.95 | 14.2 | 15.7 | -37.3 ± 9.80 | -124.89 ± 15.59 |
|  | HadGEM-RCP:8.5 | -6.82 | -5.82 | 39.34 ± 10.29 | -130.50 ± 15.27 | 7.91 | 8.69 | -33.21 ± 7.12 | -105.98 ± 13.82 |
|  | HadGEM-RCP:4.5 | -8.48 | -7.48 | -29.63 ± 7.84 | -100.68 ± 11.96 | 7.33 | 8.06 | -24.25 ± 6.43 | -82.49 ± 10.42 |
|  | HadGEM-RCP:2.6 | -8.84 | -7.83 | -28.27 ± 7.49 | -96.69 ± 11.45 | 6.99 | 7.69 | -23.12 ± 6.15 | -79.25 ± 9.89 |
|  | MRI-RCP:8.5 | -5.92 | -4.92 | -35.02 ± 10.05 | -132.11 ± 14.94 | 0.68 | 1.79 | -29.62 ± 8.50 | -111.72 ± 13.44 |
|  | MRI -RCP:4.5 | -6.27 | -5.28 | -33.31 ± 9.66 | -127.24 ± 14.21 | 0.65 | 1.63 | -28.06 ± 8.15 | -107.26 ± 12.71 |
|  | MRI-RCP:2.6 | -7.24 | -6.24 | -29.33 ± 8.50 | -113.60 ± 13.02 | 0.62 | 1.64 | -24.54 ± 7.20 | -94.94 ± 11.51 |
| Arctic Canada South | CESM-LE | -10.36 | -9.36 | -22.95 ± 6.72 | -87.2 ± 14.94 | 6.07 | 6.68 | -19.59 ± 5.74 | -74.35 ± 13.44 |
|  | HadGEM-RCP:8.5 | -12.84 | -11.81 | -27.06 ± 7.67 | -98.25 ± 15.16 | 1.71 | 1.88 | -23.22 ± 6.58 | -84.24 ± 13.54 |
|  | HadGEM-RCP:4.5 | -15.25 | -14.25 | -22.61 ± 6.61 | -85.21 ± 13.43 | 1.77 | 1.94 | -19.18 ± 5.62 | -72.29 ± 11.85 |
|  | HadGEM-RCP:2.6 | -15.98 | -14.98 | -20.70 ± 6.08 | -78.53 ± 12.35 | 1.94 | 2.14 | -17.42 ± 5.13 | -66.22 ± 10.75 |
|  | MRI-RCP:8.5 | -11.57 | -10.75 | -23.13 ± -6.51 | -85.31 ± 14.90 | 0.01 | 1.29 | -27.42 ± 7.72 | -101.32 ± 16.66 |
|  | MRI-RCP:4.5 | -13.12 | -12.11 | -20.22 ± 6.97 | -92.55 ± 13.11 | 0.01 | 1.31 | -24.06 ± 7.02 | -92.55 ± 15.44 |
|  | MRI-RCP:2.6 | -13.55 | -12.55 | -23.65 ± 6.97 | -91.87 ± 15.09 | 0.01 | 1.29 | -19.83 ± 5.92 | -77.16 ± 13.74 |
| Arctic Canada North | CESM-LE | -18.76 | -17.76 | -23.57 ± 7.78 | -96.8 ± 16.25 | 2.66 | 2.93 | -23.88 ± 7.50 | -99.66 ± 15.79 |
|  | HadGEM-RCP:8.5 | -8.08 | -7.08 | -40.33 ± 11.16 | -139.64 ± 28.63 | 16.59 | 18.25 | 25.42 ± 7.95 | -97.45 ± 25.42 |
|  | HadGEM-RCP:4.5 | -10.08 | -9.08 | -33.41 ± 9.71 | -124.43 ± 26.30 | 22.96 | 22.26 | -21.60 ± 6.31 | -80.65 ± 23.25 |
|  | HadGEM-RCP:2.6 | -11.84 | -10.84 | -23.91 ± 7.11 | -91.83 ± 23.80 | 31.07 | 34.18 | 12.39 ± 3.70 | -47.82 ± 20.72 |
|  | MRI-RCP:8.5 | -19.2 | -18.21 | -19.22 ± 5.70 | -75.04 ± 12.15 | 0.05 | 0.05 | -20.26 ± 5.98 | -78.87 ± 11.79 |
|  | MRI-RCP:4.5 | -20.12 | -19.12 | -20.32 ± 5.87 | -77.45 ± 11.55 | 0.03 | 0.03 | -22.49 ± 6.48 | -85.49 ± 12.20 |
|  | MRI-RCP:2.6 | -21.34 | -20.34 | -21.57 ± 6.19 | -81.62 ± 11.89 | 0.02 | 0.03 | -21.85 ± 6.32 | -83.38 ± 11.99 |





**Table A3.** Glaciers or ice cap with direct observations used in the GRACE constrained model with their corresponding sensitivity values used in the model calibration

| Glacier/ ice cap | Lat | Long | Area (km$^2$) | Elev (m) | Sensitivity parameters | | | | | | |
| --- | --- | --- | --- | --- | --- | --- | --- | --- | --- | --- | --- |
| | | | | | lp | $T_o$ | $K_o$ | DDF | $d_{prec}$ | $\Delta$h | $\Delta$p |
| Gulkana | 63.28 | -143.43 | 17.57 | 1858 | -0.0065 | 274.15 | 0.59 | 2.3 | 0.15 | 1600 | 1000 |
| Wolverine | 60.41 | -148.91 | 16.75 | 1257 | -0.0065 | 273.15 | 0.49 | 2.3 | 0.15 | 1500 | 1000 |
| Devon | 75.06 | -81.51 | 3388.25 | 1316 | -0.005 | 274.15 | 0.75 | 3.00 | 1.5 | 1400 | 600 |
| Meighen | 79.98 | -99.19 | 92.93 | 146 | -0.0065 | 275.15 | 0.65 | 3.00 | 0.55 | 1400 | 800 |
| White | 79.51 | -90.88 | 40.51 | 1169 | -0.0057 | 274.15 | 1.00 | 3.00 | 0.55 | 1200 | 800 |