# Peer review of "21st century estimates of mass loss rates from glaciers in the Gulf of Alaska and Canadian Archipelago using a GRACE constrained glacier model"

_The Cryosphere, 2019_

## Referee Comment (RC1) · Anonymous Referee #1 · 13 Mar 2020

Overview: This paper develops a model to simulate the mass balance of glaciers in the Gulf of Alaska and Canadian Arctic Archipelago region. It calibrates the model to GRACE gravity observations and then uses the model, forced by GCM projections, to predict future glacier changes through 2100. The simulations show greater rates of glacier mass loss than most other published studies.

The methods used here follow the Wahr, Burgess and Swenson (2016, hereafter WBS16) approach to simulate the mass balance of several mountain glacier regions. There are differences in the way GRACE data are processed, with this study using

Slepian functions, whereas WBS16 use spherical harmonics combined with fitting functions to assign mass changes to specific 0.5 degree mascons. The overall mass budget modeling approach is nearly identical. WBS16 do separate calculations for glacier versus non-glacier terrain, whereas this study appears to only focus on glacier covered areas.

General comments: There is a lack of clarity in the description of the methods in this paper. One challenge is that the terminology is similar but not identical to WBS16. For example, the authors use "M" to represent the glacier mass balance, whereas for WBS16 "M" is the total glacier mass. This creates problems later, for example in Eq. 11 where the mass balance M(t), which is by definition the change in glacier mass at time t, is incorrectly related directly to the GRACE-derived mass. Note that the GRACE solution, if it was perfectly isolated from other sources of mass change, could be correctly labeled the cumulative glacier mass balance, or the total glacier mass relative to an unknown starting mass (M_0 in WSB16) representing the glacier state at the start of the GRACE mission. However it is not correct to say the GRACE data are the same as glacier mass balance.

Another example is equation 15, equating the time evolution of glacier volume to the mass balance corrected for density, which is different from WBS16's equation used to estimate the initial state of glacier mass based on a global glacier volume inventory. Differences in these formulations relate to the time span under consideration, because an instantaneous addition or removal of glacier mass does not necessarily have the density of glacier ice. Finally, Line 223 is particularly problematic, with Delta_M labeled as a change in mass balance, which would be the second derivative of the glacier mass, and Delta_M(t) labeled as the change in mass, which is just the mass balance. The authors are recommended to stick with the WBS16 terminology to minimize confusion, or to identify their own variables these are aligned with the Glossary of Mass Balance Terms (Cogley et al., 2011).

Corrections for terrestrial water storage: note that Beamer et al., 2016

(doi.org/10.1002/2015WR018457) simulated the water budget of glacier and non-glacier terrain in the Gulf of Alaska. They show that GRACE solutions are capturing the full land + ice signal. This mirrors a similar finding for the Canadian Arctic Archipelago (Lenaerts et al., 2013, doi:10.1002/grl.50214). These studies show that even those GRACE solutions that forward model the terrestrial water balance, for example by using GLDAS data, are not capable of isolating the glacier mass budget signals alone. This means the earlier work of Arendt et al (2008, 2013) was incorrect in asserting that GRACE Gulf of Alaska signals represented just the glacier mass balance. This likely explains why your modeled time series show smaller seasonal amplitudes than the GRACE observations (e.g. Figure 1).

Comparison to field observations: The comparison between the model and conventional mass balance observations (e.g. Line 239-) is unclear. It appears the model is being tuned to a specific location, presumably the model grid closest to the centroid of the field observations? This use of a generalized regional model to predict the mass balance at a single location is advised against in WBS16. As justification for this approach, the authors point to a related analysis in Arendt et al. (2013), but that is a different method that was exploring how representative "index glaciers" were of regional mass balance patterns reflected in a 1x1 degree GRACE mascon. In other words, that was a scaling up of the area-averaged mass balance of a single glacier to a region for direct comparison with GRACE observations, with no tuning involved. It is not too surprising that a degree-day model could be sufficiently tuned to represent ground observations especially for those glaciers whose mass balances are most controled by temperature, such as Gulkana.

Specific Comments:

Line 69-70: Arendt et al. (2008) calculate 7 Gt/yr as the LIA contribution. Be sure to reference Larsen et al. (2005).

Line 70: This is the first time RGI is introduced. Be sure to spell out the acronym and

provide a reference. Also, how exactly is the mask generated, for example, does the cell need to have >50% glacier cover to be considered a glacier, or just any amount?

Line 120: RGI 6.0 describes the most updated glacier state. Do earlier RGI versions represent ice conditions closer to the start of the GRACE record? Alaska glacier areas have changed considerably in the 2002-2017 period.

Line 131: There is a problem with terminology related to glacier regions. This study appears to focus on the RGI first order region 01 ("Alaska"), but excludes the second order region 01 ("North") including the glaciers of the Brooks Range in northern Alaska. I believe the confusion arises from Harig and Simons (2016) designation of north and south Gulf of Alaska regions aimed at distinguishing between RGI first order regions 01 and 02 (Alaska and Western Canada). All of this traces back to Arendt et al.'s (2013) decision to extend the label "Gulf of Alaska" to those glaciers extending into the coast ranges of BC, since they also drain into the Gulf of Alaska. In any event, your manuscript is covering well over 20,000 glaciers in Alaska/Yukon/NW BC. I recommend revisiting the RGI subregions and using those to specify which glaciers are included here.

Figure 1: The y-axis is the cumulative mass change from an arbitrary starting mass. The label "mass change" is incorrect since this would be the first derivative of the mass and would need to be in units of mass / time.

Line 135-136: Arendt et al's (2002) work refers to differences in northern Alaska glaciers (e.g. Brooks Range) and those along the southern coast (e.g. St. Elias, Chugach mountains). Harig and Simons (2016) would be a better reference here.

Line 214: Specify which assumptions of the initial state are used, becaues the use of area-volume scaling requires estimation of the initial volume state of the glaciers.

Line 240-242: "Gulkana and Wolverine..." This sentence should include a reference to the data (O'Neel et al., 2019, doi: doi.org/10.1017/jog.2019.66).

Figure A2: Check the units on the y-axis, as thousands of Gt each year would be far too much mass change. State explictly that these are annual mass balances. A reference should be provided rather than saying "from WGMS". I suggest different colors are not needed to distinguish Alaska from Canadian Arctic glaciers.

---

## Referee Comment (RC2) · Anonymous Referee #2 · 27 Apr 2020

Ashokkumar and Harig present a regional glacier model, constrained through GRACE-based observations, for Alaska and the Canadian Arctic. Generally speaking, more diversity within the glacier model ensemble is needed, and because of this, additions are needed, timely, and of interest to the community.

However, the manuscript contains a multitude of inaccuracies in terms (e.g., frequent use of "extrapolation" when relatively complex models are meant) and references to the literature (e.g., quoting non-modeling studies as background for modeling issues). I've listed them below under specific comments and suggestions, but the high number

of these implies that a general and major revision of the manuscript is needed to get rid of them. Generally, the manuscript makes a somewhat sloppy impression (e.g., tables and figures in the appendix seem to be in an almost random order, and some are apparently not referred to in the text at all; instead, reference is made to tables that don't exists – most frequently, to table 5). In some figures, it is unclear what is shown, or at least the axis labels don't make sense (e.g., lower row of Fig. 2: this cannot be precipitation – or it is the non-corrected precipitation (eq. 13), which would not make sense to show here, either).

In Sect. 2.3, these inaccuracies lead to a very severe problem, because it is not possible to follow the reasoning behind the mass balance model equations, nor is it possible to really understand how the model is (supposed to be) working (e.g., reference to hypsometry, but then lapse rates are calculated based on the difference of regional mean glacier elevation and glacier mean elevation, etc.). There is a lot of confusion of variables. E.g., $\Delta h$ is defined both as an elevation (L151) and as a lapse rate (L172) – neither of which makes much sense to me, considering eq. 8. Overall, it is impossible for me to judge whether the model is set up in a meaningful way.

The model validation, which is central for a study like this, takes four lines in the manuscript (L 184-187) and is otherwise found in (more or less uncommented) Figures in the appendix, and somewhat spread out through the rest of the manuscript (mostly in 4.3 and 4.4). These validation results need to be presented in the main text, and need to be discussed in much more depth. Also, they should be compared to the performance of other, similar models (i.e., those that contributed to the intercomparison in Hock et al., 2019). Since the authors search parameter space for the minimum RMSE, they should also present and discuss the model's sensitivity to these parameter values. Much could be learned from such an analysis. Finally, in the validation/optimization, not even an attempt is made to measure out-of-calibration-sample performance. This is absolutely necessary to have a good estimate of how the model will be doing when not replicating known observations, but e.g. reconstructing or projecting mass loss

during periods (or in places) for which no observations exist.

Generally speaking, I get the impression that the authors do not have an adequate understanding of what the state-of-the-art is in regional glacier models. There are frequent misunderstandings of the literature (listed below) – most severely perhaps that the model presented here "intrinsically accounts for higher order dynamics" (L293-294) or is "incorporating higher order of glacier dynamics" (L334-335), which is simply not true, and implying that it is thus superior to the type of models intercompared in Hock et al. (2019). Nothing is said about ice dynamics in the manuscript, leading me to assume that dynamics are simply ignored (which none of the models in Hock et al., 2019, does). Similarly, there are frequent references to modeling papers, implying that they extrapolate in-situ observations to the regional scale (or something similar, listed below). This is a severe misrepresentation of what these models do.

Overall, the authors create the impression (willingly or not – I cannot tell) that their model approach is superior to what is typically done in state-of-the-art models. (i) The equations they present do not confirm this; (ii) the evaluation is not presented in depth, and not compared to other models' evaluations, such that it is impossible to say how the model is doing in comparison to others; (iii) there are some clear misrepresentations of what other models are doing.

I hope that my relatively strong opinion here is not misunderstood: it is important that more glacier models on regional scales are developed, and I strongly and sincerely welcome efforts to do so. I don't even think it is necessary, or even desirable, that new glacier models are more complex or more accurate than existing ones. But the description of the model, and the motivation of modeling choices, needs to be a lot clearer than is the case here, and there need to be in-depth evaluations of model performance. I cannot and don't want to rule out that the model presented here is reasonable. But there are many indications in the text and equations that it is not (listed in detail below), or the authors were "only" extremely unclear in their presentation.
Based on this assessment, I cannot recommend the manuscript for publication in The Cryosphere.

Specific comments and suggestions:

- Title/throughout the paper: you are referring to the "Gulf of Alaska" and use it as synonymous with "Alaska", the name of the RGI region (e.g., Figs. 3, 4). Is your region definition different from the RGI's? If so, the direct comparison with the Hock et al. (2019) data is problematic. If it is not different, please use the same name.

- L5: "extrapolate": I think "project" is a better word, since "extrapolate" invokes associations with simple linear extension of a time series.

- L7: "highest" compared to what? The other considered regions, or globally, or temporally? - It is unclear in the abstract what the ranges of numbers given refer to. Please indicate that they correspond to the lowest and highest ensemble member.

- L8-10: I don't see a reason for singling out ACS in the abstract: Fig. 4 shows that generally, your model is on the high mass loss side of the Hock et al. (2019) ensemble, and really sticks out in Alaska for RCP4.5, in ACN with one ensemble member in RCP4.5 and RCP8.5, and in ACS in RCP2.6.

- L17: remove dot after m in "m.w.e."

- L22: name of region missing after "-3 Gt yrˆ-2"

- L23: "has" -> "have"

- L31-33: I would argue that "extrapolation of regional mass balance from about 255 direct observations to represent 200,000 glaciers worldwide (Cogley, 2009)." is a misrepresentation of both what is done typically in models (which do not use observations to extrapolate, but as a constraint – and which therefore can produce global numbers that a very different from an extrapolation) and what is done in Cogley (2009), since he doesn't do any modeling.
- L36: "Southern" Hemisphere

- L37: Again, none of the three studies you cite "extrapolates" any mass balances. I think what you want to say is that the models are too weakly constrained, since there are not enough mass balance observations in these regions, leading to large uncertainties.

- L39: "several studies" – please cite examples

- L43-45: Unclear what is meant: what "process"? How could model parameters be able to represent mass balance? I don't think the choice in Huss & Hock (2015) to use regional observations was based on an inability of the model to produce numbers for individual glaciers (the model is based on individual glaciers).

- L50: It is neither clear what "uncertainties from extrapolation of direct observations" are meant here (see above, the models do not "extrapolate"), nor how this is relevant for "issues in volume-area scaling".

- L52-53: Unclear what is meant, since Arendt et al. (2013) don't do any modeling. (However, almost all glacier models do account for spatial variability within a region, so it's unclear why the reference is made to that paper).

- L58-59: "which perturb the geoid at a spatial resolution of several hundred kilometers" makes it sounds like the perturbation is on the scale of several hundred kilometers, but that is actually only the resolution of GRACE; please rephrase for more accuracy.

- L68: "the glacial isostatic adjustment (GIA) model" – which one? There are many around.

- Sect. 2.1 and 2.2: I'm not a GRACE expert, and don't know the technical literature well enough to really evaluate these sections. My therefore somewhat superficial impression is that this part of the manuscript is the most mature. However, I'm wondering about the motivation of producing GRACE-based estimates specifically for this manuscript: The overall goal is to project glacier mass change in the three regions

using a model that is constrained by GRACE. Why then not use a previously published GRACE-based estimate? There may be good reasons, but they are not obvious to me, and not stated in the manuscript. At the moment, while the sections read well, they take away focus from the main storyline of the manuscript.

- Fig. 1, upper row: shouldn't the color bar be labeled "mass change"? Lower row: please include a legend for the lines (which is model, which is GRACE).

- L119: please either indicate that you are here referring only to the period of model calibration, or also introduce the GCM data you are using

- L121: please use either degree day of temperature index; I think temperature index is more correct since it is more general (i.e., also applicable at monthly time scales)

- L123: delete "such as the glacier outlines, area and elevation", since this is what hypsometry means

- L124: delete "glacier"

-L131-132: you have already introduced ACN and ACS, you can use them here (or add "Arctic" in front of "Canada North").

- L131: why do you only mention the number for "Gulf of Alaska North", not Alaska (or Gulf of Alaska) entirely? That they are different from Gulf of Alaska South might motivate to consider them separately, but you could (and should) still also include the southern part.

- L 136: "From this information, we compute the area elevation distribution of glaciers at every 50 m grid spacing." How is this different from the hypsometry contained in the RGI?

- L145-146: Then why include it in eq. 6?

- L147: "at" -> "a"

[Figure]

- L147: why a degree day, if monthly time steps are used (L141)?

- L148: it's not the number of days above threshold temperature, but – as given in eq. 7, the sum of temperatures above threshold.

- Eq. 7: why isn't $T\_gl$ used here?

- Eq. 8: Why do you use mean glacier elevation when you want to consider glacier hypsometry? And why use \Delta h as a basis for the correction, shouldn't this be the elevation of the geopotential surface from which T is taken?

- L155: why only convective precipitation? Is the rest of precipitation ignored? (Also: same questions regarding h and \Delta h as for eq. 8).

- L158: The precipitation and temperature lapse rates should be VERY different, they are completely different things!

- L162: what is "snowfall from all the glaciers"? I don't understand.

- L164: "We ignore the effects of tidewater calving from Gulf of Alaska and Canadian Archipelago since they contribute less to regional mass balance (Larsen et al., 2015)." This is a very strong assumption, which would easily explain a higher temperature sensitivity of this model (i.e., mass loss from frontal ablation is treated like surface mass loss, which implies an overestimation of surface mass loss, which leads to too high degree-day factors in the calibration).

- L172-173: \Delta h is not a lapse rate (see L151), \Delta p should not be a lapse rate according to eq.9 (but it is not clear to me, what is should be); the precipitation gradient d_prec is closest to what I would understand as a lapse rate in eq. 9 (but that depends, of course, on what \Delta p actually is).

- L175ff: I think "parameter space" is meant, instead of "model space". Why choose these three parameters? Do you mean that the modeled mass balance has greater sensitivity to these parameters? Can you show this?

[Figure]

- L175: I don't think you solve for the parameters, but optimize them.

- Eq. 11: units are missing. What motivates the choices of boundaries of parameters space?

- L184-185: How do you measure explained variance? What about a model bias, the amplitude of variability, etc.? How big is the minimal RMSE that you obtain?

- Fig. A3 is referred to before Fig. A2, Fig. A1 is not referred to at all.

- Table 5 is the first table that is referred to (and where is it?).

- Figs. A4 to A6: what are "modeled observations"? How do the model results agree with the observations? This should be discussed in the text, and compared to the performance of other, similar models (e.g., those in Hock et al., 2019).

- L199-200: what does this mean: "closely modeled the climate model data"?

- L200: the delta approach (eq. 13) should remove any bias. What is meant here?

- L207-208: sentence incomplete.

- L213-214: I don't understand the reasoning of "since our model constrained by GRACE observations has secular and seasonal trends in mass balance."

- L213: Is it then correct that the model keeps the area and hypsometry of glaciers constant, even in the projections until 2100? Again, this is a severe limitation compared to other models, and would be a very simple explanation why it projects more mass loss than other models.

- L218ff: I'm not sure if I understand this correctly, but it would not make sense to re-tune the model after the application of anomalies from the GCMs (eq. 12 and 13) to maximize the agreement between model results and observations, since the GCMs will be in very different states of climate variability.

- L221: where is table 5?
- Fig. 2: What is shown in the lower row? The vertical axis label does not make sense, the different GCMs do not have differences in precipitation that differ by a factor large than 2. What are the gray lines in the middle row?

- Fig. 3: why does the middle row of panels say "CESM"? the legend implies that the different GCMs are coded in the line styles. In Fig. 4, it's in the center column.

- L229ff: Unclear whether this refers to the results when GCM data are used as boundary condition, or if not, why/how/if this is different from what is stated further above.

- L238: where is table 5? The results from the optimization need to be discussed in much more detail.

- L239-249: how do these results compare with other models' performance in these regions?

- L253-255: "Uncertainties in the volume and mass loss rates depends primarily on the (i) initial conditions of volume, (ii) glacier hypsometry or area changes, and (iii) sensitivity to temperature and (iv) precipitation." How do you arrive at this conclusion?

- Sect. 4: it is fine that you compare with previous results. However, there are some significant assumptions in you model (pointed out above) that are not addressed, and that easily would explain the increased mass losses you see (and which would be unphysical).

- L270: "Like the existing glacier models": not all of the use a temperature-index approach.

- L274: they were not extrapolated, but projected using models.

- L278: Which problem? Huss and Hock (2015) is one of the models used in Hock et al. (2019). The other models in Huss and Hock (2019) used very different strategies.

- L285ff: I'm not convinced that this is advantage, unless you show that your model's performance is actually superior to the other models' performance. I would not as-

sume so, since the higher accuracy of the regional mean comes at the cost of lower spatial resolution. The results shown in Figs. A5 to A6 let me doubt that your model's performance is clearly superior. It may be, but you need to show this.

- L293ff: "Our model constrained by GRACE intrinsically accounts for higher order dynamics": I do not understand at all how a seasonality signal should say something about dynamics. In the manuscript, there is no statement on treatment of dynamics in the model whatsoever, which leads me to the assumption that it is simply ignored. All models in Hock et al. (2019) have some simplified representation of dynamics. Figs. A5 and A6 show mass balance, and in this short time period, there will not be any discernible impact of ice dynamics.

- Fig. 5: please use region names consistently

- Fig. 5: What range is represented? Max-min, or some percentile of the ensemble? It would be useful to have information also on a central value (e.g., mean or median) of the ensemble.

- L303-end: I will not continue with as detailed comments as above, since there will be major revisions necessary, presumably changing the results considerably.

- L303-307: much more interesting than arbitrary changes to glacier geometry and temp/precip would be the discussion of the model's sensitivity to the parameters.

- L321ff: this should be shown. Based on what is presented in the manuscript, I assume that the higher rates are explained by (i) ignoring the contribution of frontal ablation to mass change in the calibration of the surface mass balance model, and (ii) in ignoring glacier geometry change in the projections.

- L334: this is simply not true. No word is said on "higher order glacier dynamics" except that it is somehow incorporated. Where and how is this the case?

- L356-357: "This method eliminates the need for extrapolation of direct observations for regional mass balance and SLE as in Radi′c et al. (2014)." Radic et al. (2014) do

not extrapolate direct observations for regional mass balance.

- L358ff: how do the units used here agree with the monthly time scale mentioned further above?

- L365: again, I don't think \Delta h in eq. 8 is a lapse rate (nor should it be a parameter that is optimized)

- Sect. 4.3. and 4.4 should go into (a new) section on model validation, presented before the results.

- Conclusion a: You have not shown this, since you have not shown such a comparison. I do not see any measurement of regional bias in the manuscript.

- Conclusion b: I don't agree that you have shown that the model's results are good enough to suggest they can replace conventional field observations, and I utterly disagree, given Fig. A5 and A6.

- Conclusion c: I disagree that you have shown "that Arctic Canada South has greater sensitivity in the recent decade, and our model is able to capture this sensitivity". The greater sensitivity is likely an artifact of modeling choice, see above.

---

## Referee Comment (RC3) · Anonymous Referee #3 · 30 Apr 2020

**Review of '21$^{st}$ century estimates of mass loss rates from glaciers in the Gulf of Alaska and Canadian Archipelago using a GRACE constrained glacier model'**
by L. Ashokkumar and C. Harig
In this manuscript, Ashokkumar and Harig present a new estimate of recent mass loss (2002-2017) based on GRACE gravimetric observations. These observations are then used to calibrate the parameters of a glacier evolution model that relies on volume-area scaling and for which the surface mass balance (SMB) is calculated from a degree-day approach. The authors compare the so-modelled mass losses with estimates from other research groups (Hock et al., 2019). For the Gulf of Alaska and for Arctic Canada North, they find mass losses that are in line with other estimates from the literature. For Arctic Canada South, their newly estimated mass loss is substantially higher than previous estimates from the literature.

The new gravimetric data presented in this paper seems sound and will likely be of interest to the glaciological community. However, the manuscript mostly focuses on using these data to calibrate a glacier evolution model and to simulate the future evolution of glaciers. There are some rather fundamental problems with this main part of the manuscript:

1. Limited novelty:
   a. The model relies on volume area scaling to simulate the future evolution of glaciers. V-A scaling was a widely used technique in regional glacier evolution studies at a time when glacier thickness distribution was largely unknown and when computational constraints did not allow for more elaborate approaches. Over recent years, several methods have been developed to estimate the ice thickness distribution over a large sample of glaciers (see e.g. Huss & Farinotti, 2012; Farinotti et al., 2019). In combination with a better characterization of glaciers and increasing computational resources, **more sophisticated methods (vs. V-A scaling) have been developed and successfully used to simulate a large ensemble of glaciers**. This includes methods in which the glacier geometry is explicitly accounted for and on which changes are imposed based on parameterizations relying on observations (e.g. Huss & Hock, 2015; Rounce et al., 2020a) and methods in which the ice dynamics are explicitly included to simulate the future evolution of glaciers (e.g. Clarke et al., 2015; Maussion et al., 2019; Zekollari et al., 2019). Some of these methods have already been applied at a global scale (Huss & Hock, 2015; Hock et al., 2019; Maussion et al., 2019; Marzeion et al., 2020). In this respect, using a V-A scaling approach at a regional scale is far from being novel and may even be considered to be a bit outdated... The authors claim that their model is able to account for 'higher-order dynamics', which is really not the case with a V-A scaling approach (nor is it with any of the other large-scale glacier evolution models available from the literature).
   b. Also the **climatic conditions** used in this study are slightly outdated. The authors rely on ERA-Interim data, while now a more sophisticated and higher-resolution product is available: ERA5 (see https://www.ecmwf.int/en/forecasts/datasets/reanalysis-datasets/era5).
   Also for the future simulations, it is a pity that the authors relied on CMIP5 GCM output and not on CMIP6 GCM output. In fact, given the regional focus

of their study, the most logical option would have been to rely on regional climate model (RCM) output from the 'North American CORDEX Program' (https://na-cordex.org/)

c. The authors compare their results to the output of the **GlacierMIP** (Hock et al., 2019). The problem with GlacierMIP is the fact that the simulations are difficult to compare given the large difference occurring not only in terms of glacier model, but also in terms of boundary conditions (different glacier volume, different forcing,...etc.). This has now in part been solved in **GlacierMIP2** (Marzeion et al., 2020), which is the first coordinated experiment for glacier evolution modelling. I am well aware that this is a brand-new study and that the authors could not have been aware of this. It is therefore not really a point of critique, but I think that it would nevertheless be good if the comparison could be made with these new results.

d. **Calibrating model parameters with regional values is tricky in general** (reasoning behind this is elaborated in bullet point 2). The best method to calibrate a large-scale glacier evolution model is to perform a calibration of model parameters at the glacier level by reproducing glacier-specific observations. Glacier specific mass change estimates are now becoming widely available (e.g. Brun et al., 2017; Braun et al., 2019; Dussaillant et al., 2019; Shean et al., 2020), and have been used to calibrate regional glacier evolution models (e.g. Zekollari et al., 2019; Rounce et al., 2020a, 2020b). In this sense, relying on regional estimates of glacier mass change, even if these are probably more accurate than previous regional estimates (which seems to be the case with the new GRACE data you present), is not ideal (see bullet point 3).

2. Calibrating parameters at regional scale is always quite tricky. It was done until recently, but now the priority is to transition to glacier-specific observations when calibrating models. The main reasons for this are that (i) individual glaciers within a region are subject to different mass changes and that (ii) no single combination of model parameters (to be applied for all glaciers within a region) can accurately describe this. In general, there are two options when working with regional mass changes for model calibration:

a. One can **assume that all glaciers in a given region are subject to the same mass balance and then perform a calibration of the model parameters at the glacier scale** (i.e. different parameters for every glacier). As you explain in your manuscript, this is not ideal, as in reality large glaciers tend to have a more negative mass balance (more out of balance compared to climatic conditions), compared to smaller glaciers. This is the method used by Huss and Hock (2015) and it has the advantage that you can match the mass changes for every glacier. But as you match a mass balance that is 'off' at the individual glacier level, you tend to underestimate future volume losses (as the mass balance of the large glaciers, which make up for most of the volume, is positively biased when assuming that all glaciers have the same mass balance) (e.g. Zekollari et al., 2019).

b. The second option is to **match the regional mass changes by using a single set of parameters for all glaciers within a region**. This is the option you opt for by matching the new GRACE derived mass changes with your model. For this, you need to rely on the same model parameters for all glaciers, and in an ideal case

you would be able to match regional loss (calibration) and the mass balance at the individual glacier level (evaluation). However, Huss and Hock (2015) have shown that this does not work well, as by working with the same parameters for every glacier you can get very strange mass balances at the individual glacier level: i.e. the sum of the mass balances match the regional mass balance, but the reason for this is wrong. Through this, it is likely that you have very strange mass balances at the individual glacier level: e.g. some glaciers may even have a very positive mass balance for the observational period. When you then project these in the future, these glaciers may even be growing... It would be good to have an overview of how the mass balance looks like at the individual glacier level:

- Comparison to direct mass balance observations. You perform this, but while doing you still modify some parameters (which you can't do!). Even then, the match is not very good so far.
- It would be good to describe and show the mass balances that you obtain for all glaciers (e.g. through box plot). I hope this is not too bad, but I am afraid that some glaciers may have a very strange mass balance.

3. So what is the role of GRACE? You present it like a major advantage of that you match the GRACE derived regional mass change. OK, this estimate is probably better than relying on a very rough approach in which mass balance observations from a few glaciers are used to derive a regional mass balance, **but in the end this is just a number for an entire region, which is a strong limitation for regional glacier modelling.** The difference between relying on these GRACE observations and rougher regional estimates is relatively small in the end. If a regional mass change estimate is used, I would advise you to use another approach for the calibration (bullet point 2a). And when doing so, make sure that you match the GRACE observations (see next bullet point)

4. You do not match the GRACE derived mass balance!
   a. While allowing to have a wide range of values for your parameters, you are not able to reproduce the mass balance derived from glaciers. This is quite concerning. This may be solved by changing the calibration setup (bullet point 2a).
   b. By allowing the model parameters to take any value, one can match the GRACE derived mass balance (at the regional scale). If this would have been done, the projected future glacier changes would likely have been very close to the values from the literature. In the result you present:
      - Gulf of Alaska: is the region with the best match between GRACE and modelled MB → modelled future glacier evolution relatively close to estimates from literature
      - For Arctic Canada North: the modelled MB is more negative than GRACE observations → modelled future glacier loss is higher than the estimates from the literature
      - For Arctic Canada South: modelled MB is far more negative than in GRACE observations → modelled future glacier loss is much higher than the estimates from the literature

By matching the observed MB (from GRACE), your projected mass changes will be close to the other projections from the literature. This is not surprising, as the GRACE derived mass balance is not that different from previous estimates based on the extrapolation of field measurements and/or remote observations.

I think it is a brave effort of the authors to 'jump' into a research field (glacier evolution modelling) that is new from them. It is such efforts and new impulses that will help a research field – in this case the field of glacier evolution modelling - forward. When doing so, one must first gain a good overview of this field in order to avoid making some basic conceptual mistakes and to ensure that the work is an added value. I think the authors should have focused on this, rather than pointing at wrong reasons for explaining discrepancies between their results and results from the literature. A correction based on comments above will help tackling some problems (rethink the calibration, make sure to match the GRACE observations, use state-of-the-art climate output), but even then, the novelty of results put forward will be very questionable given the model architecture. In GlacierMIP and GlacierMIP2 models of the same complexity and more sophisticated models have been used at larger regional scales.

**References**

Braun, M. H., Malz, P., Sommer, C., Farias-Barahona, D., Sauter, T., Casassa, G., et al. (2019). Constraining glacier elevation and mass changes in South America. *Nature Climate Change*, *9*, 130–136. https://doi.org/10.1038/s41558-018-0375-7

Brun, F., Berthier, E., Wagnon, P., Kääb, A., & Treichler, D. (2017). A spatially resolved estimate of High Mountain Asia glacier mass balances from 2000 to 2016. *Nature Geoscience*, *10*, 668–673. https://doi.org/10.1038/NGEO2999

Clarke, G. K. C., Jarosch, A. H., Anslow, F. S., Radić, V., & Menounos, B. (2015). Projected deglaciation of western Canada in the twenty-first century. *Nature Geoscience*, *8*, 372–377. https://doi.org/10.1038/ngeo2407

Dussaillant, I., Berthier, E., Brun, F., Masiokas, M., Hugonnet, R., Favier, V., et al. (2019). Two decades of glacier mass loss along the Andes. *Nature Geoscience*. https://doi.org/10.1038/s41561-019-0432-5

Farinotti, D., Huss, M., Fürst, J. J., Landmann, J., Machguth, H., Maussion, F., & Pandit, A. (2019). A consensus estimate for the ice thickness distribution of all glaciers on Earth. *Nature Geoscience*. https://doi.org/10.1038/s41561-019-0300-3

Hock, R., Bliss, A., Marzeion, B., Giesen, R. H., Hirabayashi, Y., Huss, M., et al. (2019). GlacierMIP – A model intercomparison of global-scale glacier mass-balance models and projections. *Journal of Glaciology*. https://doi.org/10.1017/jog.2019.22

Huss, M., & Farinotti, D. (2012). Distributed ice thickness and volume of all glaciers around the globe. *Journal of Geophysical Research: Earth Surface*, *117*(4), F04010. https://doi.org/10.1029/2012JF002523

Huss, M., & Hock, R. (2015). A new model for global glacier change and sea-level rise. *Frontiers in Earth Science*, *3*, 1–22. https://doi.org/10.3389/feart.2015.00054

Marzeion, B., Hock, R., Anderson, B., Bliss, A., Champollion, N., Fujita, K., et al. (2020). Partitioning the Uncertainty of Ensemble Projections of Global Glacier Mass Change. *Earth's Future*, accepted. https://doi.org/10.1029/2019EF001470

Maussion, F., Butenko, A., Champollion, N., Dusch, M., Eis, J., Fourteau, K., et al. (2019). The

Open Global Glacier Model ( OGGM ) v1.1. *Geoscientific Model Development*, *12*, 909–931. https://doi.org/10.5194/gmd-12-909-2019

Rounce, D., Hock, R., & Shean, D. (2020a). Glacier mass change in High Mountain Asia through 2100 using the open-source Python Glacier Evolution Model (PyGEM). *Frontiers in Earth Science*, *7*, 331. https://doi.org/10.3389/feart.2019.00331

Rounce, D. R., Khurana, T., Short, M., Hock, R., Shean, D., & Brinkerhoff, D. J. (2020b). Quantifying parameter uncertainty in a large-scale glacier evolution model with a Bayesian model: Application to High Mountain Asia. *Journal of Glaciology*.

Shean, D. E., Bhushan, S., Montesano, P., Rounce, D. R., Arendt, A., & Osmanoglu, B. (2020). A Systematic, Regional Assessment of High Mountain Asia Glacier Mass Balance. *Frontiers in Earth Science*, *7*(February). https://doi.org/10.3389/feart.2019.00363

Zekollari, H., Huss, M., & Farinotti, D. (2019). Modelling the future evolution of glaciers in the European Alps under the EURO-CORDEX RCM ensemble. *The Cryosphere*, *13*, 1125–1146. https://doi.org/10.5194/tc-13-1125-2019

---

## Referee Comment (RC4) · Anonymous Referee #4 · 5 May 2020

Dear authors, For time reasons, I have only looked at the GRACE processing part of the manuscript "21st century estimates of mass loss rates from glaciers in the Gulf of Alaska and Canadian Archipelago using a GRACE constrained glacier model" by Lavanya Ashokkumar and Christopher Harig. The authors use localized base functions to determine glacier mass balances from GRACE. This approach has been shown to facilitate a slightly higher spatial resolution compared to estimates based on global spherical harmonics alone. It is quite well established and was previously applied to determine mass changes in Greenland. Below are some comments that I would like to

[Figure]

see resolved in the revision of the paper.

A major concern are the treatment of signal overlaps between regions, particularly important for estimates of North Arctic Canada: could there be an influence of mass changes in Greenland on the glacier estimates? Looking at Fig. 1, considerable amount signal overlaps exist.

Another concern that would need more explanation is the calibration of the degree-day scheme of the glacier models with the GRACE data. As I understand, misfit is calculated as difference of monthly model estimates and GRACE observations. How sensitive are the optimal calibration parameters to the time scale considered for calibration? Would calibration and projection yield different results if, e.g. mainly trends or multi-year temporal components were considered?

Detailed comments

L64: Sun et al. 2016 showed that including the sea-level equation in the estimation of the geocenter motion affects recovered mass trends and annual amplitudes over the ice sheets. You are dealing with much smaller spatial scales (less impacted by low-degree harmonics). But have you checked the effect? Using Sun et al. 2016 is the recommendation of the SDS centers.

L66: Similarly to c20 (and c30) – have you checked the sensitivity to the coefficients replaced? SDS recommend Loomis et al. 2019, https://doi.org/10.1029/2019GL082929

L73: Please clarify at what stage the hydrological signals are removed – subtracted from the final mass time series or as a correction on the gravity field coefficients? Depending on the omission error and the response of your inversion approach, this could make a significant difference.

L73: Which temporal components are removed? Full signal? Only trends? Please comment on the reliability of trends and the spread of different hydrological models.

L76: "and typical inversion strategy" seems out of place. Is it the "gravity field determi-

nation approach employed"?

L83: Please provide another sentence how exactly the approach of Wahr et al. 2006 is adopted. I assume you are deriving RMS residuals from the GRACE coefficient time series? Or are you calibrating formal errors provided with the GRACE L2 data?

L114: Cut-off degree L=60 seems to be an arbitrary choice, eventually defining your Shannon number, hence spatial resolution. Why not use L=90 provided by the SDS center and truncate according to significance?

L117: What are "clear estimates of noise uncertainty"? Please specify. In addition, coefficient uncertainties are only part of the total uncertainties relevant for the mass time series. How do you deal with uncertainties related to the choice of your inversion approach (e.g. internal and external leakage), or parameters used in it? An easy way to do this is to compare to e.g. CSR mascons.

Fig. 1 and text: How do you deal with signal correlations / leakage into the region of interest?

Fig. 2, Panels a – should the label be "Total mass loss (Gt)", not "Total mass loss rate". b – "C" needs a degree symbol (you could take it for example from the legend of Figure 3 in "°K" (ïĄŁ). c – should this be Precipitation rate (mm/year) or similar?

Fig. 2, Panel 2a, there is a strange black line at the beginning of the time series. Please check.

Fig. 2 and Fig. 4 are never referenced in the text, I think. In addition, there is inconsistency with regard to using capitals and abbreviation for the text figure. . .

References

Sun, Y., Riva, R. & Ditmar, P. Optimizing estimates of annual variations and trends in geocenter motion and J 2 from a combination of GRACE data and geophysical models. J. Geophys. Res. Solid Earth 121, 8352–8370 (2016).

---

## Author Comment (AC1) · 2 Jun 2020

Dear Reviewer,

Thank you for the comments on our manuscript, *"21st century estimates of mass loss rates from glaciers in the Gulf of Alaska and Canadian Archipelago using a GRACE constrained glacier model"*. We have highlighted the referee comments in italics and our response in regular font. Here we have provided a summary of our response to the

comments stated by the reviewers.

First, there is a reference to the methods section for the lack of clarity in representation of glacier mass. This is an issue due to the terminology that was used for representing the glacier mass loss from the model and GRACE observation. This comment is also mentioned by the referee 2. We will be providing a detailed methodology and proper use of the glaciological terms that is consistent with the glacier modeling communities. Second, there is a minor issue related to the use of terrestrial water storage. The comment was mentioned by the reviewer 4 (Line 73, page C2). We will provide a clarification of how the hydrological signals from GLDAS is capable of resolving mass loss trends from gravity signals. Third, comparsion to the field or local observations using a regional glacier model. This is also one of the important question raised by the reviewer 3 (point 2b). We will be providing a section on the model validation to indicate how well the local observations are represented by our glacier model.

1. Referee: *The methods used here follow the Wahr, Burgess and Swenson (2016, hereafter WBS16) approach to simulate the mass balance of several mountain glacier regions. There are differences in the way GRACE data are processed, with this study using Slepian functions, whereas WBS16 use spherical harmonics combined with fitting functions to assign mass changes to specific 0.5 degree mascons. The overall mass budget modeling approach is nearly identical. WBS16 do separate calculations for glacier versus non-glacier terrain, whereas this study appears to only focus on glacier covered areas.*

**Slepian based processing technique**: There is a difference in the GRACE processing method used by WBS16 and the current paper for obtaining the regional mass balance. In the WBS16 technique, mass loss is obtained from small defined regions called mascons that spans 100 km$^2$, roughly 0.5 degree grid resolution. The mascon formulation, developed by the JPL, utilizes a 2 deg spherical cap to resolve the

spherical harmonics from the GRACE K-band inter-satellite range rate observations into gravity signals, accounting for full Stokes noise covariance (Luthcke et al., 2008; Ivins et al., 2011). The gravity signals are resolved for each grid and it accounts for both glaciers and non-glacierized sources within a mascon. The non-glacierized components are eliminated from the glacier signals by using terrestial water storage (TWS) and other non-anthropogenic water sources using Community Land Model. In contrast, slepian basis function has been successfully implemented for smaller spatial regions such as the Gulf of Alaska (North and South) (Harig et al., 2016). We will provide detailed methods on how we seperate glacier and non-glacier sources from GRACE gravity signals in the methods section.

2. Referee: *Another example is equation 15, equating the time evolution of glacier volume to the mass balance corrected for density, which is different from WBS16's equation used to estimate the initial state of glacier mass based on a global glacier volume inventory. Differences in these formulations relate to the time span under consideration, because an instantaneous addition or removal of glacier mass does not necessarily have the density of glacier ice. Finally, Line 223 is particularly problematic, with $\Delta M$ labeled as a change in mass balance, which would be the second derivative of the glacier mass, and $\Delta M(t)$ labeled as the change in mass, which is just the mass balance. The authors are recommended to stick with the WBS16 terminology to minimize confusion, or to identify their own variables these are aligned with the Glossary of Mass Balance Terms (Cogley et al., 2011).*

We agree that the equation 15 was incorrect, as it should have represented the initial state of glaciers. We will provide an explanation of how we derive mass loss rates from modeled glaciers in the revised submission.

3. Referee: *Corrections for terrestrial water storage: note that Beamer et al.,*

*2016 (doi.org/10.1002/-2015WR018457) simulated the water budget of glacier and nonglacier terrain in the Gulf of Alaska. They show that GRACE solutions are capturing the full land + ice signal. This mirrors a similar finding for the Canadian Arctic Archipelago (Lenaerts et al., 2013, doi:10.1002/grl.50214). These studies show that even those GRACE solutions that forward model the terrestrial water balance, for example by using GLDAS data, are not capable of isolating the glacier mass budget signals alone. This means the earlier work of Arendt et al (2008, 2013) was incorrect in asserting that GRACE Gulf of Alaska signals represented just the glacier mass balance. This likely explains why your modeled time series show smaller seasonal amplitudes than the GRACE observations (e.g. Figure 1).*

It is true that our modeled mass balance does not represent the seasonal amplitudes compared to GRACE observations. This is because our model is based on temperature indexed degree day, where the model inputs are based on ERA-Interim temperature and precitation. As mentioned in the reviewer comments 4, we will provide a detailed methodology on how we recover mass loss trends from GRACE gravity signals.

Regarding the seasonal amplitudes in the modeled mass loss, we will be updating our model based on ERA5 temperature and precipitation data products. Then, we will optimize the modeled estimates with GRACE mass loss trends. Our results will change based on this revision.

**Specific comments**
4. *Line 69-70: Arendt et al. (2008) calculate 7 Gt/yr as the LIA contribution. Be sure to reference Larsen et al. (2005).*

We agree that the LIA correction for Alaska is 7 Gt/yr and it will be included with reference in the manuscript.

5. *Line 120: RGI 6.0 describes the most updated glacier state. Do earlier RGI versions represent ice conditions closer to the start of the GRACE record? Alaska glacier areas have changed considerably in the 2002-2017 period.*

The RGI version 6.0 inventory has been used in this study to understand the glacier geometry and dynamic response due to area-elevation feedback. We agree that the glaciers in Alaska has undergone a rapid change between 2002 and 2017, but the inventories prior to version 3.0 are mostly incomplete, since ∼48 - 58% of global glacier outlines were not included (Pfeffer et al, 2014). When the extrapolation to future projections was attempted with incomplete inventory, it can lead to large uncertainties. Hock et al (2019) has made a comparsion with different version of inventory and how it impacted the future mass loss and SLE rates. Therefore, we followed the guidelines for glacier models in the GlacierMIP project to use the RGI version 6.0 in our model calibration phase (**?**).

6. *Line 131: There is a problem with terminology related to glacier regions. This study appears to focus on the RGI first order region 01 ("Alaska"), but excludes the second order region 01 ("North") including the glaciers of the Brooks Range in northern Alaska. I believe the confusion arises from Harig and Simons (2016) designation of north and south Gulf of Alaska regions aimed at distinguishing between RGI first order regions 01 and 02 (Alaska and Western Canada). All of this traces back to Arendt et al.'s (2013) decision to extend the label "Gulf of Alaska" to those glaciers extending into the coast ranges of BC, since they also drain into the Gulf of Alaska. In any event, your manuscript is covering well over 20,000 glaciers in Alaska/Yukon/NW BC. I recommend revisiting the RGI subregions and using those to specify which glaciers are included here.*

We are willing to check with RGI regions and sub-regions for Gulf of Alaska and use it in accordance with region specification set by the GlacierMIP project

All other minor corrections such as figure title, reference to appropriate papers and other typos will be revised in the manuscript submission.

References

Arendt, A.A., Luthcke, S.B., Larsen, C.F., Abdalati, W., Krabill, W.B. and Beedle, M.J., 2008. Validation of high-resolution GRACE mascon estimates of glacier mass changes in the St Elias Mountains, Alaska, USA, using aircraft laser altimetry. Journal of Glaciology, 54(188), pp.778-787.

Harig, C. and Simons, F.J., 2016. Ice mass loss in Greenland, the Gulf of Alaska, and the Canadian Archipelago: Seasonal cycles and decadal trends. Geophysical Research Letters, 43(7), pp.3150-3159.

Ivins, E.R., Watkins, M.M., Yuan, D.N., Dietrich, R., Casassa, G. and Rülke, A., 2011. On‐land ice loss and glacial isostatic adjustment at the Drake Passage: 2003–2009. Journal of Geophysical Research: Solid Earth, 116(B2).

Hock, R., Bliss, A., Marzeion, B., Giesen, R.H., Hirabayashi, Y., Huss, M., Radić, V. and Slangen, A.B., 2019. GlacierMIP–A model intercomparison of global-scale glacier mass-balance models and projections. Journal of Glaciology, 65(251), pp.453-467.

Larsen, C.F., Motyka, R.J., Freymueller, J.T., Echelmeyer, K.A. and Ivins, E.R., 2005. Rapid viscoelastic uplift in southeast Alaska caused by post-Little Ice Age glacial retreat. Earth and Planetary Science Letters, 237(3-4), pp.548-560.

Luthcke, S.B., Arendt, A.A., Rowlands, D.D., McCarthy, J.J. and Larsen, C.F., 2008. Recent glacier mass changes in the Gulf of Alaska region from GRACE mascon solutions. Journal of Glaciology, 54(188), pp.767-777.

Pfeffer, W.T., Arendt, A.A., Bliss, A., Bolch, T., Cogley, J.G., Gardner, A.S., Hagen, J.O., Hock, R., Kaser, G., Kienholz, C. and Miles, E.S., 2014. The Randolph Glacier Inventory: a globally complete inventory of glaciers. Journal of Glaciology, 60(221), pp.537-552.
* * *

---

## Author Comment (AC2) · 2 Jun 2020

We would like to thank the referee for detailed comments regarding the glacier model. In this short summary, we will discuss the major points addressed by the referee and revise them in our submission.

We are aware that our glacier model is not one of the sophicated or complex models as in GlacierMIP2. The primary objective of our model is to understand the present and future mass loss rates by model calibration using GRACE $monthly$ observations, in

contrast to single regional mass balance from geodetic observations. Also, we believe that the referee has not understood the concept of using GRACE $monthly$ observations in model calibration, instead the referee thinks that our model is calibrated with single mass balance estimate from gravimetry instead of geodetic observation as in the Huss and Hock, 2015 (Page C3 and C5 last paragraph). The major difference in our glacier model stems from the way in which the glacier area feedback is performed. While, we have used the volume area scaling compared to the advanced flowline or glacier thickness based area-evolution. It should be mentioned that 5 out of 6 models in GlacierMIP, 5 out of 11 models in GlacierMIP2 have used volume-area or volume-length or volume-area-length scaling to account for glacier geometry change.

We are planning to work on the model calibration and validation with suggestions from all reviewers, hence it is likely that some of the results and discussion will change in the revised submission. The minor comments from page C4-C10 will be addressed in the detailed submission.

1. **Use of terminologies and reference to tables/figures**: The term 'extrapolation' (instead of projection) has been used in the context of estimating the future mass loss and sea-level rates. We agree with the reviewer's comments that it is not a basic linear extrapolation, rather it is a temperature indexed glacier model constrained by GRACE monthly observations. We will revise the manuscript to read appropriate modeling terminologies, according to the standards of GlacierMIP and GlacierMIP2. We agree that the table 1 was incorrectly mentioned as table 5. And, the y-axis label in Figure 1 and Figure A5 has been corrected (comments from referee 1). In Figure 2, we have shown the temperature and precipitation after bias correction which are used as model inputs for future projections. As you may notice, there is large bias from precipitation (even after bias correction) from the GCM. This figure is similar to representation of temperature and precipitation from GCM in Figure 4 in Radic et al., 2014.

2. **Section 2.3: Inaccuracy of the modeling terminologies**: We agree that some of the formula used is confusing to understand the model setup and processing. We like to clarify that we followed the glacier mass terminologies and formula for model setup as in Wahr et al., 2016. The equation 8 and 9 represent the downscaled temperature and precitation at elevation bins, instead of temperature or precipitation at glacier. Further, it was a typo (L151) that we mentioned $h$ and $\Delta h$ as average elevation of glaciers. We are willing to address the terminology issue in the revised submission.

3. **Model validation** (Page C2): For model validation, our model was able to represent direct observations of mass balance from individual glaciers (Figure A4 - A6 in the supplementary). We would like to clarify that the L184 - 187 refer to model calibration step, where the glacier model is optimized with GRACE monthly observations. In the revised manuscript, we will be including a section on model validation as it was one of the suggestion by referee 3 (point 2b).

4. **Higher order dynamics** (Page C3, paragraph 1): We agree that we have not incorporated higher order of dynamics in our glacier models like some of the models in GlacierMIP and GlacierMIP2 (Hock et al., 2019; Marzeion et al., 2020). In the revised submission, we will make sure to exclude the term 'higher order of dynamics' and use approriate term for volume-area scaling feedback.

5. **Figure 2**: The precitation rates from different GCM are different (lower panel). We will analyse the GCM precipitation again and check if this is incorrect.

**Conclusion a:** The regional bias between the observed mass balance (GRACE) and modelled mass balance is shown in Figure 1. The comparsion is shown in the form of mass balance time series.

[Figure]

**Conclusion b:** The intension of this study is to calibrate glacier models with GRACE *monthly* observations, in contrast to GlacierMIP where the model calibration was based on **single** regional estimate of mass balance. Here we are not trying to re-place direct observations of mass balance, instead our model calibrated from GRACE monthly observations does not require any in-situ observations in model calibration. This is the very first attempt in glacier modelling community to test a model without inputs from direct observations. Figure A5 and A6 indicate the modelled rates of mass balance from individual glaciers and the agreement between modelled and measured direct mass balance.

**Conclusion c:** For the sentence "The Arctic Canada South has greater sensitivity of mass balance rates..". In the revised submission, we will be attempting the model calibration with inputs from ERA5 and optimization of parameters and we will revise the conclusions about the higher sensitivity in the ACS.

References

Hock, R., Bliss, A., Marzeion, B., Giesen, R.H., Hirabayashi, Y., Huss, M., Radić, V. and Slangen, A.B., 2019. GlacierMIP–A model intercomparison of global-scale glacier mass-balance models and projections. Journal of Glaciology, 65(251), pp.453-467.

Huss, M. and Hock, R., 2015. A new model for global glacier change and sea-level rise. Frontiers in Earth Science, 3, p.54.

Marzeion, B., Hock, R., Anderson, B., Bliss, A., Champollion, N., Fujita, K., Huss, M., Immerzeel, W., Kraaijenbrink, P., Malles, J.H. and Maussion, F., 2020. Partitioning the Uncertainty of Ensemble Projections of Global Glacier Mass Change. Earth's Future, p.e2019EF001470.

Radić, V., Bliss, A., Beedlow, A.C., Hock, R., Miles, E. and Cogley, J.G., 2014. Regional and global projections of twenty-first century glacier mass changes in response to climate scenarios from global climate models. Climate Dynamics, 42(1-2), pp.37-58.

Wahr, J., Burgess, E. and Swenson, S., 2016. Using GRACE and climate model simulations to predict mass loss of Alaskan glaciers through 2100. Journal of Glaciology, 62(234), pp.623-639.

---

## Author Comment (AC3) · 2 Jun 2020

We would like to thank the reviewer for this constructive review. In this discussion comment we would like to broadly address and share our thoughts on these comments. We look forward to addressing these comments in detail in a revision of our manuscript.

Generally, in reference to the novelty of our glacier model, it was not our intention to create the most state of the art or complex glacier model. Our work's novelty stems

from the use of GRACE $monthly$ observations as our regional constraint. In particular, we choose to produce our own GRACE mass balance estimates because we constrain our glacier model at the monthly time resolution of GRACE. We do not use the multi-year trend of mass change, but 163 months from 2002-2017. Because of this, we chose to use modeling procedures common to several other models so that we could compare our results with theirs and examine the impact of using GRACE data as a model constraint.

1a. **Use of volume area (V-A) scaling**: Yes there are more sophisticated methods than the V-A scaling. We chose to use V-A scaling because we wish to directly compare our results against other models, and examine the impact of using GRACE data. 5 of the 6 models in the GlacierMIP paper (Hock et al., 2019 table 1) and 5 of 11 models in the GlacierMIP2 paper (Table 1 in Marzeion et al., 2020) use some form of volume-area scaling relation (v-a, v-a-l, v-a-l-response time, etc.). We think V-A scaling is preferable for the ease of comparison to prior work.

1b. **Use of ERA-I**: Our use of ERA-Interim was partly due to ERA5 not being available at the project start, and partly due to improve comparison to prior work. Since submission we have migrated to ERA5 and we can easily update these results to ERA5 in our revision.

**Use of CMIP5 results**: We use CMIP5 GCM output specifically because it covers the same time period as our GRACE observations. CMIP6 output only begins in 2015, and therefore would not allow us to solve for the bias parameters necessary during 2002-2017. It also appears the equilibrium climate sensitivity of CMIP6 models is quite different from CMIP5, and this would likely make a comparison to prior glacier models very challenging.

1c. **Comparsion with GlacierMIP2**: We are aware of the recent publication in glacier model, GlacierMIP2. It must be noted that some of the models are regionally focussed such as the AND2012, GloGEMflow, KRA2017 and PyGEM, which leaves us to include GLIMB, JULES and OGGM in the comparsion (Anderson et al., 2012; Kraaijenbrink et al., 2017; Maussion et al., 2020; Rounce et al., 2020; Shannon et al., 2019; Zekollari et al., 2019). We will try our best to use the same set of conditions in the model comparsion such as the initial volume, boundary conditions in the revised manuscript.

1d. see point 2

2. **Regional parameter calibration**: We have matched the regional mass balance with a single set of parameters for each region (option 2b), and as a result we would expect that any individual glacier within the region would likely have poor agreement between model and observed mass balances. Our purpose in using monthly GRACE data is that it is very different and independent from what is used in other models. (By this we mean glacier specific mass balance estimates which are perhaps at best annualized) We believe that we could, in the future, use the spatial information in GRACE with our glacier model to improve performance. And perhaps the way forward on that is to ascribe the regional mass balance to each individual glacier such that it is consistent with the broad spatial pattern from GRACE. However, as we mention in point 3, this is a much more complex process which we have reserved for future work.

3. **Use of GRACE data**: We do not use a single number (such as the trend) from GRACE to constrain our glacier model. Instead we use all the 163 monthly observations between Jan 2002 and June 2017, which capture the seasonal processes, as constraints in our model. In-situ mass balance observations are typically limited to annual temporal resolution. Individual glacier mass balance estimates from DEMs are typically annualized from multi-year DEM differencing (e.g. Shean et al., 2020).

Additionally, we believe there is additional information in the GRACE data that could be used to improve our model in future work. For example, currently we constrain our model on the total integrated mass of the region. In future work, we plan to apply this to the timeseries of each Slepian basis function, which will allow us to use the spatial information from GRACE within the region to constrain our model. Our first step in this paper is to show the initial performance with GRACE data.

4. **Matching GRACE derived mass balance**: We agree that this is a weakness in our manuscript. We believe part of the poor agreement is due to the poor representation of precipitation in ERA products. When we re-run the model results using ERA5 input fields, we are very willing to examine this disagreement further by e.g. testing an even wider range of parameter space.

References

Anderson, B. and Mackintosh, A., 2012. Controls on mass balance sensitivity of maritime glaciers in the Southern Alps, New Zealand: the role of debris cover. Journal of Geophysical Research: Earth Surface, 117(F1).

Kraaijenbrink, P.D.A., Bierkens, M.F.P., Lutz, A.F. and Immerzeel, W.W., 2017. Impact of a global temperature rise of 1.5 degrees Celsius on Asia's glaciers. Nature, 549(7671), pp.257-260.

Marzeion, B., Hock, R., Anderson, B., Bliss, A., Champollion, N., Fujita, K., Huss, M., Immerzeel, W., Kraaijenbrink, P., Malles, J.H. and Maussion, F., 2020. Partitioning the Uncertainty of Ensemble Projections of Global Glacier Mass Change. Earth's Future, p.e2019EF001470.

Maussion, F., Butenko, A., Champollion, N., Dusch, M., Eis, J., Fourteau, K., Gregor,

P., Jarosch, A.H., Landmann, J., Oesterle, F. and Recinos, B., 2019. The Open Global Glacier Model (OGGM) v1. 1. Geoscientific Model Development, 12(3), pp.909-931.

Rounce, D.R., Khurana, T., Short, M.B., Hock, R., Shean, D.E. and Brinkerhoff, D.J., 2020. Quantifying parameter uncertainty in a large-scale glacier evolution model using Bayesian inference: application to High Mountain Asia. Journal of Glaciology, 66(256), pp.175-187.

Shannon, S., Smith, R., Wiltshire, A., Payne, T., Huss, M., Betts, R., Caesar, J., Koutroulis, A., Jones, D. and Harrison, S., 2019. Global glacier volume projections under high-end climate change scenarios. The Cryosphere, 13, pp.325-350.

Shean, D.E., Bhushan, S., Montesano, P., Rounce, D.R., Arendt, A. and Osmanoglu, B., 2020. A Systematic, Regional Assessment of High Mountain Asia Glacier Mass Balance. Front. Earth Sci, 7, p.363.

Zekollari, H., Huss, M. and Farinotti, D., 2019. Modelling the future evolution of glaciers in the European Alps under the EURO-CORDEX RCM ensemble. The Cryosphere, 13(4), pp.1125-1146.

---

## Author Comment (AC4) · 2 Jun 2020

We would like to thank the reviewer for constructive comments about the GRACE processing. In this discussion comment, we would like to broadly address and share our thoughts on these comments, and we look forward to addressing them in detail in the manuscript revision.

The major comments from the reviewer are related to the GRACE processing and its influence on the model sensitivities. We agree that there are issues related to signal overlap when recovering mass loss from Arctic Canada North, which is spatially adjacent to Greenland. We will also address the errors related to the use of degree one coefficients and C20, suggested by the reviewers. Further, we will provide a clear methodology in the GRACE processing (section 2.1) on how the GRACE signals are recovered for the sea level calculations. One of the interesting analysis has been the calibration of DDF and other model parameters based on the multi-year trend in the GRACE time series.

1. *A major concern are the treatment of signal overlaps between regions, particularly important for estimates of North Arctic Canada: could there be an influence of mass changes in Greenland on the glacier estimates? Looking at Fig. 1, considerable amount signal overlaps exist.*

From our prior work in Harig et al., 2015; 2016, there can be a signal leakage when the slepian functions are recovered from smaller spatial regions, such as the Arctic Canada North when there is a large concentration of gravity anomaly, that is from Greenland. Slepian functions are known to recover for effective signal recovery, in terms of spatial (smaller regions) and spectral resolution from the number of bandwidth. We confirm that the signal leakage is not double counted in the GRACE mass loss estimation. A detailed explaination for signal recovery and issues related to leakage will be demostrated in the form of synthetic experiments as in Von hippel and Harig, 2019 and it will included in the Appendix of revised paper.

2. *Another concern that would need more explanation is the calibration of the degree-day scheme of the glacier models with the GRACE data. As I understand, misfit is calculated as difference of monthly model estimates and GRACE observations. How sensitive are the optimal calibration parameters to the time scale considered for calibration? Would calibration and projection yield different results if, e.g. mainly trends or*

*multi-year temporal components were considered?*

This is an interesting observation about the multi-year temporal trend and we thank the reviewer for this question. We expect that there could be model sensitivities depending on the time period used in the model calibration. For example, if we were to consider the time period after 2010, mass loss from the three region are higher since 2012, and therefore it will influence the future loss rates based on the calibration period. However, if we were to consider a time period prior to 2008, say 2002 - 2007, it can have a different uncertainty on the future mass loss rates due to lower mass loss rates compared to post 2012 period. We are also likely to have different values for model parameters, say the degree day factor (DDF) or threshold temperature. We plan to do a sensitivity test on a time scale, as it can provide information about DDF and how GRACE contribute to model sensitivity (Wouters et al., 2019).

The main reason for considering the mass loss during 2002 and 2017 in the model calibration, is to account for all the inter-annual and annual seasonalities from GRACE solutions and also to match our model calibration with other glacier model studies in the GlacierMIP. We look forward to addressing the model sensitivities for multi-year trend in the revised submission.

3. *L64: Sun et al. 2016 showed that including the sea-level equation in the estimation of the geocenter motion affects recovered mass trends and annual amplitudes over the ice sheets. You are dealing with much smaller spatial scales (less impacted by lowdegree harmonics). But have you checked the effect? Using Sun et al. 2016 is the recommendation of the SDS centers.*

In our manuscript, we have used the version of degree 1 coefficients based on Swenson et al., 2008 (processed on June 2019) and we noticed that the degree 1 coefficients has not incorporated the effects of atmosphere and ocean dealiasing (AOD). We will record the difference in GRACE mass trend according to the recommendation by the

reviewer.

4. *L66: Similarly to c20 (and c30) – have you checked the sensitivity to the coefficients replaced? SDS recommend Loomis et al. 2019, https://doi.org/10.1029/2019GL082929.*

Our GRACE processing is based on C20 coefficients, described in the technical note 11 (Cheng et al., 2013). We will update the C20 with the technical note 13 and assess the trend changes in Alaska and Canadian Archipalego.

5. *L73: Please clarify at what stage the hydrological signals are removed – subtracted from the final mass time series or as a correction on the gravity field coefficients? Depending on the omission error and the response of your inversion approach, this could make a significant difference. Which temporal components are removed? Full signal? Only trends? Please comment on the reliability of trends and the spread of different hydrological models.*

We subtract the hydrological model prior to estimating the mass loss time series. The mass loss trends are recovered from full signals. Currently, there are several hydrological models available for comparsion. However, we used a hydrological model by GLDAS as in Gardner et al., 2013 that is in accordance with the glaciological communities.

6. **GRACE processing**: In the section 2.1 (Lines 76,83,114,117), there has been few minor questions about the GRACE inversion strategy, adopted from Wahr et al., 2006 and how the uncertainties from inversion are dealt compared to CSR mascons.

In the revised manuscript, we will provide more details about the inversion approach used and detailed description of noise uncertainties will be included.

The other minor corrections in the figures will also be revised in the manuscript.

References

Cheng, M., Tapley, B.D. and Ries, J.C., 2013. Deceleration in the Earth's oblateness. Journal of Geophysical Research: Solid Earth, 118(2), pp.740-747.

Gardner, A.S., Moholdt, G., Cogley, J.G., Wouters, B., Arendt, A.A., Wahr, J., Berthier, E., Hock, R., Pfeffer, W.T., Kaser, G. and Ligtenberg, S.R., 2013. A reconciled estimate of glacier contributions to sea level rise: 2003 to 2009. science, 340(6134), pp.852-857.

Harig, C. and Simons, F.J., 2015. Accelerated West Antarctic ice mass loss continues to outpace East Antarctic gains. Earth and Planetary Science Letters, 415, pp.134-141.

Harig, C. and Simons, F.J., 2016. Ice mass loss in Greenland, the Gulf of Alaska, and the Canadian Archipelago: Seasonal cycles and decadal trends. Geophysical Research Letters, 43(7), pp.3150-3159.

Swenson, S., Chambers, D. and Wahr, J., 2008. Estimating geocenter variations from a combination of GRACE and ocean model output. Journal of Geophysical Research: Solid Earth, 113(B8).

von Hippel, M. and Harig, C., 2019. Long-term and inter-annual mass changes in the Iceland ice cap determined from GRACE gravity.

Wouters, B., Gardner, A.S. and Moholdt, G., 2019. Global glacier mass loss during the GRACE satellite mission (2002-2016). Frontiers in earth science, 7.

---

## Author Comment (AC5) · 2 Jun 2020

We would like to thank the reviewer for detailed comments regarding the glacier model. In this short summary, we will discuss the major points addressed by the review. We believe the issues brought up can be addressed in a revision to the satisfaction of the reviewer. Several of the review points arise from a miscommunication on our part in describing our model. Guided by all of the reviews, we plan to carefully edit the manuscript for clarity.

[Figure]

We are aware that our glacier model is not one of the more complex models as some of those in GlacierMIP2. The primary novelty of our model is its use of GRACE *monthly* observations as calibration data, in contrast to single regional mass balance trend estimates from other geodetic observations. While Huss and Hock., 2015 calibrate regional models with a single trend estimate for each region, we use 163 monthly GRACE observations which therefore include seasonal components and year-to-year variability. We have not made this point clear enough, as it seems to have caused some confusion to the reader (see e.g. page C5 last paragraph). We plan to address this in our revision.

The major difference in our glacier model to others stems from the way in which the glacier area feedback is performed. We have used volume-area scaling compared to the advanced flowline or glacier thickness based area-evolution used in some other models. We note that 5 out of 6 models in GlacierMIP and 5 out of 11 models in GlacierMIP2 have used volume-area or volume-length or volume-area-length scaling to account for glacier geometry change.

We are planning to work on the model calibration and validation with suggestions from all reviewers, hence it is likely that some of the results and discussion will change in the revised submission. The minor comments from page C4-C10 will be addressed in the detailed submission.

1. **Use of terminologies and reference to tables/figures**: The term extrapolation has been used in several instances, and we will rewrite/clarify these uses to properly characterize the constraints on the complex glacier models. It was our intent to describe the limitations of using a small number of in-situ mass balance estimates as model constraints for a large region, but we will rewrite these instances. We will revise the manuscript to use more appropriate modeling terminologies according to the standards of GlacierMIP and GlacierMIP2. We agree that the table 1 was incorrectly mentioned as table 5. And, the y-axis label in Figure 1 and Figure A5 has been corrected (com-

ments from referee 1). In Figure 2, we have shown the temperature and precipitation after bias correction which are used as model inputs for future projections. As you may notice, there is large bias from precipitation (even after bias correction) from the GCM. This figure is similar to representation of temperature and precipitation from GCM in Figure 4 in Radic et al., 2014.

2. **Section 2.3: Inaccuracy of the modeling terminologies**: We agree that some of the formula used is confusing to understand the model setup and processing. We were primarily following the terminologies and formula for model setup as in Wahr et al., 2016. The equation 8 and 9 represent the downscaled temperature and precipitation at elevation bins, instead of temperature or precipitation at glacier. Further, it was a typo (L151) that we mentioned $h$ and $\Delta h$ as average elevation of glaciers. We plan a careful edit of the manuscript for accuracy and clarity in our revision.

3. **Model validation** (Page C2): For model validation, we will be including a section on model validation in our revision that examines the distribution of modeled mass balances over the glacier population. It was also a suggestion by referee 3 (point 2b). In contrary to the models in GlacierMIP1, we have not considered direct observations in the calibration step, which enables us to compare our model for individual glacier mass balance. As pointed out by the referee 3, we would like to examine the model validation to understand the performance of all individual glaciers in a region. We would like to clarify that the L184 - 187 refer to model calibration step, where the glacier model is optimized with GRACE monthly observations.

4. **Higher order dynamics** (Page C3, paragraph 1): We agree that we have not incorporated higher order of dynamics in our glacier models like some of the models in GlacierMIP and GlacierMIP2 (Hock et al., 2019; Marzeion et al., 2020). Here our intent was to describe how our model includes constraints on the seasonal components

and inter-annual variability. In the revised submission, we will remove these terms and clarify our descriptions of GRACE constraints.

5. **Figure 2**: The precipitation rates from different GCMs are shown (lower panel). We will analyze the GCM precipitation again and check if this is incorrect.

**Conclusion a:** The regional bias between the observed mass balance (GRACE) and modelled mass balance is shown in Figure 1 (curves are offset). The comparsion is shown in the form of mass balance time series.

**Conclusion b:** The intent of this study is to calibrate glacier models with GRACE $monthly$ observations, in contrast to GlacierMIP where the model calibration was based on **single** regional estimate of mass balance. Here we are not trying to replace direct observations of mass balance, instead our model calibrated from GRACE monthly observations does not require any in-situ observations in model calibration.

**Conclusion c:** For the sentence "The Arctic Canada South has greater sensitivity of mass balance rates..". In the revised submission, we will perform the model calibration with inputs from ERA5 and optimization of parameters and we will revise the conclusions about the higher sensitivity in the ACS.

References

Hock, R., Bliss, A., Marzeion, B., Giesen, R.H., Hirabayashi, Y., Huss, M., Radić, V. and Slangen, A.B., 2019. GlacierMIP–A model intercomparison of global-scale glacier mass-balance models and projections. Journal of Glaciology, 65(251), pp.453-467.

Huss, M. and Hock, R., 2015. A new model for global glacier change and sea-level rise. Frontiers in Earth Science, 3, p.54.

Marzeion, B., Hock, R., Anderson, B., Bliss, A., Champollion, N., Fujita, K., Huss, M., Immerzeel, W., Kraaijenbrink, P., Malles, J.H. and Maussion, F., 2020. Partitioning the Uncertainty of Ensemble Projections of Global Glacier Mass Change. Earth's Future, p.e2019EF001470.

Radić, V., Bliss, A., Beedlow, A.C., Hock, R., Miles, E. and Cogley, J.G., 2014. Regional and global projections of twenty-first century glacier mass changes in response to climate scenarios from global climate models. Climate Dynamics, 42(1-2), pp.37-58.

Wahr, J., Burgess, E. and Swenson, S., 2016. Using GRACE and climate model simulations to predict mass loss of Alaskan glaciers through 2100. Journal of Glaciology, 62(234), pp.623-639.